# Boosting Search Engines with Interactive Agents

**Leonard Adolphs**[†*]                                                   *leonard.adolphs@inf.ethz.ch*

**Benjamin Boerschinger**[‡]                                             *bboerschinger@google.com*

**Christian Buck**[‡]                                                       *cbuck@google.com*

**Michelle Chen Huebscher**[‡]                                          *michellechen@google.com*

**Massimiliano Ciaramita**[‡]                                             *massi@google.com*

**Lasse Espeholt**[‡]                                                     *lespeholt@google.com*

**Thomas Hofmann**[†]                                                  *thomas.hofmann@inf.ethz.ch*

**Yannic Kilcher**[†*]                                                 *yannic.kilcher@inf.ethz.ch*

**Sascha Rothe**[‡]                                                       *rothe@google.com*

**Pier Giuseppe Sessa**[†*]                                              *sessap@ethz.ch*

**Lierni Sestorain Saralegui**[‡]                                         *lierni@google.com*

[†]*ETH, Zurich* [‡]*Google Research*

**Reviewed on OpenReview:** *https://openreview.net/forum?id=0ZbPmmB61g*

## Abstract

This paper presents first successful steps in designing search agents that learn meta-strategies for iterative query refinement in information-seeking tasks. Our approach uses machine reading to guide the selection of refinement terms from aggregated search results. Agents are then empowered with simple but effective search operators to exert fine-grained and transparent control over queries and search results. We develop a novel way of generating synthetic search sessions, which leverages the power of transformer-based language models through (self-)supervised learning. We also present a reinforcement learning agent with dynamically constrained actions that learns interactive search strategies from scratch. Our search agents obtain retrieval and answer quality performance comparable to recent neural methods, using only a traditional term-based BM25 ranking function and interpretable discrete reranking and filtering actions.

## 1 Introduction

Can machines learn to use a search engine as an interactive tool for finding information? Web search is the portal to a vast ecosystem of general and specialized knowledge, designed to support humans in their effort to seek relevant information and make well-informed decisions. Utilizing search as a tool is intuitive, and most users quickly learn interactive search strategies characterized by sequential reasoning, exploration, and synthesis (Hearst, 2009; Rutter et al., 2015; Russell, 2019). The success of web search relies on machines learning human notions of relevance, but also on the users' ability to (re-)formulate appropriate

---

[*]Work carried out in part during internships at Google.

queries, grounded in a tacit understanding of strengths and limitations of search engines. Given recent breakthroughs in language models (LM) (Vaswani et al., 2017; Devlin et al., 2019; Brown et al., 2020) as well as in reinforcement learning (RL) (Mnih et al., 2013; Silver et al., 2016; Berner et al., 2019), it seems timely to ask *whether*, and *how*, agents can be trained to interactively use search engines. However, the lack of expert search sessions puts supervised learning out of reach, and RL is often ineffective in complex natural language understanding (NLU) tasks. The feasibility of autonomous search agents hence remains an open question, which inspires our research.

We pursue a design philosophy in which search agents operate in structured action spaces defined as generative grammars, resulting in compositional, productive, and semantically transparent policies. Further domain knowledge is included through the use of well-known models and algorithms from NLU and information retrieval (IR). Most notably, we develop a self-supervised learning scheme for generating high-quality search session data, by exploiting insights from relevance feedback (Rocchio, 1971), used to train a supervised LM search agent based on T5 (Raffel et al., 2020). We also build an RL search agent based on MuZero (Schrittwieser et al., 2020) and BERT (Devlin et al., 2019), which performs planning via rule-constrained Monte Carlo tree search and a learned dynamics model.

We run experiments on an open-domain question answering task, OpenQA (Lee et al., 2019). Search agents learn diverse policies leading to deep, effective explorations of the search results. The MuZero agent outperforms a BM25 (Robertson & Zaragoza, 2009) search function running over a Wikipedia index, on both retrieval and answer quality metrics. This result provides novel evidence for the potential of knowledge-infused RL in hard NLU tasks. The T5 agent can more easily leverage large pre-trained encoder-decoders and proves superior to MuZero. Furthermore, a straightforward ensemble of agents is comparable in performance to the current reference neural retrieval system, DPR (Karpukhin et al., 2020), while relying solely on interpretable, symbolic retrieval operations. This suggests new challenges for future work; e.g., involving hybrid architectures and policy synthesis. We open-source the code and trained checkpoints for both agents.[1,2]

## 2 Learning to Search

It has been a powerful vision for more than 20 years to design search engines that are intuitive and simple to use. Despite their remarkable success, search engines are not perfect and may not yield the most relevant result(s) in one shot. This is particularly true for rare and intrinsically difficult queries, which may require interactive exploration by the user to be answered correctly and exhaustively.

It can be difficult for users to formulate effective queries because of the information gap that triggers the search process in the first place (Belkin et al., 1982). O'Day & Jeffries (1993) found that reusing search results content for further search and exploration is a systematic behavior (aka "orienteering"), a key ingredient for solving the information need. Lau & Horvitz (1999) analyzed a dataset of one million queries from the logs of the Excite search engine and report an average session length of 3.27 queries per informational goal. Teevan et al. (2004) noticed that users facing hard queries can even decide to partially by-pass the search engine by issuing a more general query and then navigating the links within the returned documents to find an answer. Downey et al. (2008) observed that a user's initial query is typically either too specific or too general and the amount of work required to optimize it depends on the query frequency, with infrequent queries requiring longer search sessions. They estimate from logs that tail information needs require more than 4 queries, while common ones require less than 2 (on average). Contextual query refinement is a common technique (Jansen et al., 2009), even among children (Rutter et al., 2015), used to improve search by combining evidence from previous results and background knowledge (Huang & Efthimiadis, 2009). Such refinements often rely on inspecting result snippets and titles or on skimming the content of top-ranked documents. This process is iterative and may be repeated until (optimistically) a satisfactory answer is found.

It seems natural to envision artificial search agents that mimic this interactive process by learning the basic step of generating a follow-up query from previous queries and their search results while keeping track of the best results found along the way. We call this the *learning to search* problem.

---

[1] `https://github.com/google-research/google-research/tree/master/muzero`

[2] `https://github.com/google-research/language/tree/master/language/search_agents`

### 2.1 Search Engine and Query Operations

We make the assumption that agents interact with a search engine operating on an inverted index architecture (Croft et al., 2009, §2.2), which is popular in commercial engines and IR research. Specifically, we use Lucene's implementation[3] as the search engine, in combination with the BM25 ranking function (Robertson & Zaragoza, 2009). We frame search as the process of generating a sequence of queries $q_0, q_1, \ldots, q_T$,[4] where $q_0$ is the initial query, and $q_T$ is the final query – where the process stops. Each query $q_t$ is submitted to the search engine to retrieve a list of ranked documents $\mathcal{D}_t$.

We focus on the case where $q_{t+1}$ is obtained from $q_t$ through *augmentation*. A query may be refined by adding a keyword $w \in \Sigma^{\mathrm{idx}}$, such that $q_{t+1} = q_t w$, where $\Sigma^{\mathrm{idx}}$ is the vocabulary of terms in the search index. The new term will be interpreted with the usual disjunctive search engine semantics. Furthermore, a query can be augmented by means of *search operators*. We concentrate on three unary operators: '+', which limits results to documents that contain a specific term, '-' which excludes results that contain the term, and '$\wedge_i$' which boosts a term weight in the BM25 score computation by a factor $i \in \mathbb{R}$. In addition, the operator effect is limited to a specific document *field*, either the content or the title. As an example, the query 'who is the green guy from sesame street' could be augmented with the term '+contents:muppet', which would limit the results returned to documents containing the term 'muppet' in the body of the document.

Only a small fraction of users' queries include search operators, and this behavior is not well studied. Croft et al. (2009, §6.2) estimate that less than 0.5% use '+'. However, it is noteworthy how power users can leverage dedicated search operators, in combination with sophisticated investigative strategies, to solve deep search puzzles (Russell, 2019). Additionally, unary operators are associated with explicit, transparent semantics and their effect can be analyzed and interpreted. Crucially, however, as we show in this paper, these operators are also pivotal in designing effective search agents because they allow us to generate self-supervised search session training data in a principled fashion.

### 2.2 Results Aggregation and Observations Structure

Web searchers expect the best answer to be among the top few hits on the first results page (Hearst, 2009, §5) and pay marginal attention to the bottom half of the *10 blue links* (Granka et al., 2004; Joachims et al., 2005; Nielsen & Pernice, 2009; Strzelecki, 2020). Likewise, a search agent considers only the top $k$ documents returned by the search engine at every step; we set $k = 5$ in all our experiments.

During a search session the agent maintains a list of the top-$k$ documents overall, which is returned at the end as the output. To aggregate the results from different steps during the search session we use a Passage Scorer (PS) which builds upon a pre-trained BERT model. For each result document $d \in \mathcal{D}_t$, the PS component estimates the probability of $d$ containing the (unspecified) answer $\mathrm{P}(d \ni \textsc{answer} \,|\, q) \in [0; 1]$. This probability can be viewed as a score that induces a calibrated ranking across all result documents within a session. Notice that the score is always computed conditioning on the original query $q = q_0$ and not $q_t$.

At each session step a search agent computes a structured *observation* representing the state of the session up to that point. The observation includes the query tokens and refinements describing $q_t$. The top-$k$ documents in the session are represented by their title and a text snippet. The snippet is a fixed-length token sequence centered around the text span that contains the most likely answer for $q$, as predicted by a Machine Reader (MR) (Rajpurkar et al., 2016). For ranking (PS) and answer span prediction (MR) tasks we use the same BERT system as in (Karpukhin et al., 2020). Query and aggregated results yield a segmented observation token sequence $o_t$ which is truncated to length $\leq 512$, a common input length for pre-trained transformer-based LMs (cf. Appendix B for more details and examples).

---

[3] https://lucene.apache.org/.

[4] We also refer to the query sequence as a session, or search episode.

[5] The answer is "Oscar the Grouch" who is a green *muppet* that lives in a *trash* can on Sesame street.

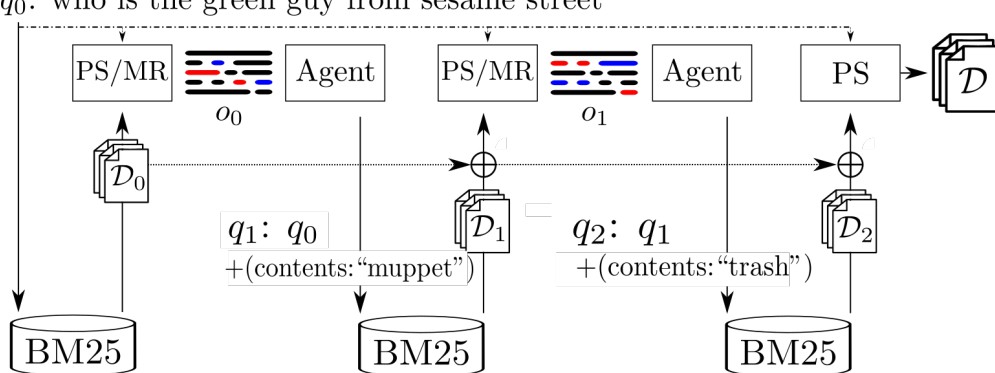

Figure 1: Schematic agent interaction with the search engine (BM25) for the query "who is the green guy from sesame street".[5]This is a real example from the query expansion procedure described in Section 2.3, see also Table A.9 for an expanded version. After receiving an initial set of documents ($\mathcal{D}_0$) for the original question, the corresponding observation ($o_0$) is compiled by ranking the documents according to their Passage Score (PS), and creating snippets for the top-$k$ documents around the answers extracted by the Machine Reader (MR). Note that PS/MR always conditions on $q_0$. The first action of the agent is to enforce the term "muppet" to be in the content of the search results. The new document set $\mathcal{D}_1$ is returned by the search engine and aggregated with the previous documents. Again, the set of documents is ranked by the Passage Scorer, and the subsequent observation for the agent is compiled. The agent then chooses to enforce the presence of the topical term "trash" and obtains another set of documents that are, again, aggregated and scored. The final result $\mathcal{D}$ contains the top-$k$ documents observed during the episode, according to the Passage Score.

The next step involves a language model which produces an embedding $\mathbf{s}_t$ from which the search agent will generate the next query. We can represent diagrammatically the operations that lead to a query refinement as

$$
\begin{bmatrix} q_0, \ldots, q_t \\ \text{search} \downarrow \text{engine} \\ \mathcal{D}_0, \ldots, \mathcal{D}_t \end{bmatrix} \underbrace{\xmapsto{\text{MR/PS}} o_t}_{\text{observation}} \underbrace{\xmapsto{\text{LM}} \mathbf{s}_t}_{\text{encoding}} \underbrace{\xmapsto{\text{agent}} q_{t+1}}_{\text{generation}} \tag{1}
$$

At each step $t$ the top-$k$ documents in the session are identified by means of their PS score. An observation $o_t$ is computed for the top documents by means of a machine reader (MR). Then the search agent's LM encodes the observation $o_t$ and decodes the next query $q_{t+1}$. Figure 1 illustrates the search agent and its components at work with an example.

## 2.3 Rocchio Query Expansions and Rocchio Sessions

The query operations introduced above allow us to generate synthetic search sessions in a self-supervised manner, making use of question-answer pairs $(q, a)$. We initialize $q_0 = q$ and aim to find a sequence of refinements that make progress towards identifying high-quality documents, based on a designed scoring function which combines retrieval and question answering performance (cf. Eq. 7, introduced in §4). A query is *not* further refined if no score increasing refinement can be found or a maximal length is reached.

To create candidate refinements, we put to use the insights behind relevance feedback as suggested in Rocchio (1971). Formalizing the query operations introduced in Section 2.1, an elementary refinement – called a *Rocchio expansion* – takes the form

$$
q_{t+1} := q_t \, \Delta q_t, \Delta q_t := [ + | - | \wedge_i \; \text{TITLE} | \text{CONTENT} ] \; w_t, w_t \in \Sigma_t := \Sigma_t^q \cup \Sigma_t^\tau \cup \Sigma_t^\alpha \cup \Sigma_t^\beta \tag{2}
$$

where $i$ is the boosting coefficient and $\Sigma_t$ refers to a set of terms accessible to the agent. By that we mean terms that occur in the observation $o_t$ – the search state at time $t$. We use superscripts to refer to the vocabularies induced from the observation which identify the terms occurring in the question ($q$), titles ($\tau$),

Table 1: An observed example Rocchio session for the question "who won season 2 great british baking show". The colored span is the answer span prediction of the machine reader, indicating if the answer is wrong (red) or correct (blue). The top BM25 retrieval results for the original query are passages from the articles about "The Great American Baking Show" – the American version of the show mentioned in the query. The reason for this confusion is that the British show is called "The Great British *Bake Off*", while the query term "baking" matches the title of the wrong document. The first Rocchio expansion *boosts* the term "final", i.e., puts more weight on this term while computing the relevance score. This is a reasonable choice as the terms is likely related to the culmination of a periodic event, such as a seasonal show. In the two subsequent steps the procedure requires the terms "bake" and "2" to be contained in the title of the retrieved documents. In this way the results first continue to shift from the American Baking Show to the British Bake Off, and eventually settle on the desired British Bake Off (series 2). The composite IR and QA score (defined in Eq.7) increases from 0.040 for the original query to 0.552 for the final query.

| | **Query and Search Results** | | **Score** |
|---|---|---|---|
| $q_0$ | who won season 2 great british baking show | | |
| | Top-2 documents retrieved with $q_0$: | | 0.040 |
| $d_1$ | Title | The Great American Baking Show | |
| | Content | . . . The first two seasons were hosted by Nia Vardalos and Ian Gomez, with **Mary Berry** from the original "GBBO" series and . . . | |
| $d_2$ | Title | The Great American Baking Show (season 3) | |
| | Content | . . . , ABC announced that **Vallery Lomas** won the competition, beating out runners-up Cindy Maliniak and Molly Brodak in the **final** week . . . | |
| $q_1$ | who won season 2 great british baking show (contents:"final"∧8) | | |
| | Top-2 documents retrieved with $q_1$: | | 0.142 |
| $d_1$ | Title | The Great British Bake Off | |
| | Content | . . . The finalists were Brendan Lynch, James Morton and **John Whaite**, the last of whom won the final in a surprise result. . . . | |
| $d_2$ | Title | The Great American Baking Show (season 2) | |
| | Content | . . . In the final technical, **Mary Berry** set the challenge on the bakers to create a British Battenberg cake with a checkerboard . . . | |
| $q_2$ | who won season 2 great british baking show (contents:"final"∧8) +(title:"bake") | | |
| | Top-2 documents retrieved with $q_2$: | | 0.186 |
| $d_1$ | Title | The Great British Bake Off | |
| | Content | . . . The finalists were Brendan Lynch, James Morton and **John Whaite**, the last of whom won the final in a surprise result. . . . | |
| $d_2$ | Title | The Great British Bake Off | |
| | Content | . . . The final of the series where **John Whaite** was crowned the winner saw its highest . . . | |
| $q_3$ | who won season 2 great british baking show (contents:"final"∧8) +(title:"bake") +(title:"2") | | |
| | Top-2 documents retrieved with $q_3$: | | 0.552 |
| $d_1$ | Title | The Great British Bake Off (series 2) | |
| | Content | . . . The competition was won by **Joanne Wheatley**. There was no Star Baker this week, as Paul and Mary felt . . . | |
| $d_2$ | Title | The Great British Bake Off (series 2) | |
| | Content | . . . contestants went on to a career in baking or have a change of career as a result of appearing on the show. **Joanne Wheatley** has written two best selling books on baking . . . | |

predicted answer spans ($\alpha$) or bodies ($\beta$) of documents in $o_t$. Note that adding terms $\notin \Sigma_t$ would make refinements more difficult to reproduce for an agent and thus would provide supervision of low utility.

A crucial aspect of creating search sessions training data based on Rocchio expansions has to do with the search complexity of finding optimal sequences of such expansions. The success of this search relies on the notion of relevance feedback. We introduce $q_* = q + a$ as the "ideal" query: query $q$ executed on the subset of documents that contain answer $a$. The results of $q_*$ define the vocabulary $\Sigma_*$. We can now define two special dictionaries that will allow us to narrow down the candidate terms to appear in the next refinement

$$\Sigma_t^\uparrow = \Sigma_t \cap \Sigma_*, \quad \Sigma_t^\downarrow = \Sigma_t - \Sigma_* . \tag{3}$$

During the search for an optimal session, it is possible to use accessible terms $w_t$ as additional keywords, or in combination with exact match ('+'), or weight boosting ('$\wedge$'), if they also occur in the ideal result set ($w_t \in \Sigma_t^\uparrow$); and to exclude $w_t$ ('-') if they are not present in the ideal results ($w_t \in \Sigma_t^\downarrow$). As in Rocchio algorithm, this is meant to bring the query closer to the relevant documents and farther away from the irrelevant ones. We have found experimentally that this leads to a good trade-off between the quality of Rocchio expansions and the search effort to find them. We call a sequence of Rocchio expansions a *Rocchio session*. Table 1 illustrates a Rocchio session for the query 'who won season 2 great british baking show', based on the experimental setup described in Section 5.

The search for Rocchio sessions is done heuristically. Full implementation details with pseudo-code illustrating the procedure and examples can be found in §5, Appendix A, and Appendix G – cf. also Table A.10.

## 3    Search Agents

### 3.1    Self-Supervised T5 Agent

It is straightforward to train a generative search agent in a supervised manner on the Rocchio sessions. We use T5, a popular pretrained transformer encoder-decoder model. As a search agent, T5 learns to predict a new search expansion from each observed state. In the spirit of *everything-is-string-prediction*, state and expansions are represented as plain strings. See Appendix B for a full example.

Our T5 agent is trained via Behavioral Cloning (BC) (Michie, 1990). We treat each step in a Rocchio session as a single training example. As is common in sequence prediction tasks, we use the cross-entropy loss for optimization. BC is perhaps the simplest form of Imitation Learning (IL), and has been proven effective in a variety of application domains (Sharma et al., 2018; Rodríguez-Hernandez et al., 2019). In our query refinement task, it allows to inherit the expressive power of the Rocchio query expansions and, differently from other IL approaches (Ross et al., 2011; Ho & Ermon, 2016; Dong et al., 2020), requires only *offline* interactions with the search engine. Crucially, this enables scaling to the large action spaces and model sizes typical of recent LMs. Our T5 agent can also be described as a Decision Transformer with fixed max return (Chen et al., 2021).

### 3.2    Reinforcement Learning: MuZero Agent

Learning to search lends itself naturally to be modeled as a reinforcement learning problem. To explore also the feasibility of learning search policies from scratch, we implement an RL search agent based on MuZero (Schrittwieser et al., 2020). MuZero is a state-of-the-art agent characterized by a learnable model of the environment dynamics. This allows the use of Monte Carlo tree search (MCTS) to predict the next action, in the absence of an explicit simulator. In our use case, MuZero aims to anticipate the latent state implied by each action with regard to the results obtained by the search engine. For instance, in the example of Figure 1, it may learn to predict the effect of using the term 'muppet' in combination with a unary operator. This approach to planning is intuitive for search, as searchers learn to anticipate the effect of query refinements while not being able to predict specific results. Furthermore, this offers a performance advantage of many orders of magnitude against executing queries with the *real* search engine.

### 3.2.1 Grammar-Guided Search

To map observations to states, the MuZero agent employs a custom BERT with dedicated embedding layers to represent the different parts (cf. Appendix B for details). Compared to T5, MuZero has a more challenging starting point: its BERT-based representation function is pre-trained on less data, it has fewer parameters (110M vs. 11B) and no cross-attention: predictions are conditioned on a single vector, [CLS]. Moreover, it cannot as easily exploit supervised signals. However, it can more openly explore the space of policies, e.g. independent of the Rocchio expansions. Through many design iterations, we have identified it to be crucial to structure the action space of the MuZero agent and constrain admissible actions and refinement terms dynamically based on context. This provides a domain-informed inductive bias that increases the statistical efficiency of learning a policy via RL.

We take inspiration from generative, specifically context-free, grammars (CFGs) (Chomsky, 1956) and encode the structured action space as a set of production rules, which will be selected in (fixed) top-down, left-to-right order. A query refinement is generated, in a way that mimics Rocchio expansions, as follows

$$Q \Rightarrow U\,Q \,|\, W\,Q, \quad U \Rightarrow \mathrm{Op\ Field}\ W, \quad \mathrm{Op} \Rightarrow +\,|-|\wedge_i, \quad \mathrm{Field} \Rightarrow \mathrm{TITLE}\,|\,\mathrm{CONTENT} \tag{4}$$

which allows for adding plain or structured keywords using unary operators. The selection of each refinement term $W$ proceeds in three steps, the first two can be described by the rules

$$W \Rightarrow W_t^q \,|\, W_t^\tau \,|\, W_t^\beta \,|\, W_t^\alpha \,|\, W^{\mathrm{idx}}, \quad W_t^x \Rightarrow w \in \Sigma_\tau^x, \quad x \in \{q, \tau, \beta, \alpha\}, \; W^{\mathrm{idx}} \Rightarrow w \in \Sigma^{\mathrm{idx}} \tag{5}$$

which means that the agent first decides on the origin of the refinement term, i.e., the query or the different parts of the top-scored result documents, and afterwards selects the term from the corresponding vocabulary. As the term origin correlates strongly with its usefulness as a refinement term, this allows to narrow down the action space effectively. The agent is forced to pick a term from the larger vocabulary (approximately 1M terms) of the search index $\Sigma^{\mathrm{idx}}$ during MCTS, as there is no observable context to constrain the vocabulary.

The third level in the action hierarchy concerns the selection of the terms. We have found it advantageous to make use of subword units; specifically, BERT's 30k lexical rules involving word pieces, to generate terms sequentially, starting from a term prefix and adding one or more suffixes. Note that this part of the generation is context-sensitive, as we restrict node expansions to words present in the vocabulary. We make use of tries to efficiently represent each $\Sigma_\tau^x$ and amortize computation. The grammar-guided MCTS is explained in more detail in Appendix F.

## 4 The OpenQA Environment

We evaluate search agents in the context of open-domain question answering (Open-QA) (Voorhees, 2000; Chen et al., 2017). Given a question $q$, we seek documents $\mathcal{D}$ that contain the answer $a$ using a search engine, the environment. Following common practice, we use Lucene-BM25 with default settings on the English Wikipedia. BM25 has provided the reference probabilistic IR benchmark for decades (Robertson & Zaragoza, 2009), only recently outperformed by neural models (Lee et al., 2019). The Lucene system provides search operators comparable to commercial search engines.

Exploration-based learning is vulnerable to discovering adversarial behaviors. As a safeguard we design a composite reward. The score of a results set $\mathcal{D}$, given $q$, interpolates three components. The first is the Normalized Discounted Cumulative Gain (NDCG) at $k$. See Eq. 6a, where $w_i = \log_2(i+1)^{-1} / \sum_{j=1}^k \log_2(j+1)^{-1}$ are normalizing weights, and $\mathrm{rel}(d|q) = 1$, if $a \in d, 0$ otherwise:

$$a)\ \mathrm{NDCG}_k(\mathcal{D}|q) = \sum_{i=1}^k w_i\,\mathrm{rel}(d_i|q), \qquad b)\ \mathrm{NDCEM}_k(\mathcal{D}|q) = \sum_{i=1}^k w_i\,\mathrm{em}(d_i|q). \tag{6}$$

NDCG is a popular metric in IR as it accounts for rank position, it is comparable across queries, and it is effective at discriminating ranking functions (Wang et al., 2013). NDCG alone can have drawbacks: on "easy" questions a score of 1 can be achieved in short meritless episodes, while on "hard" ones it may be impossible to

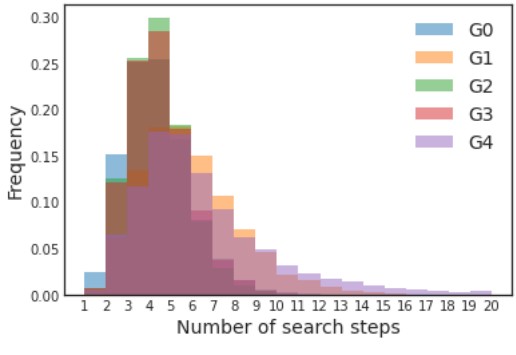
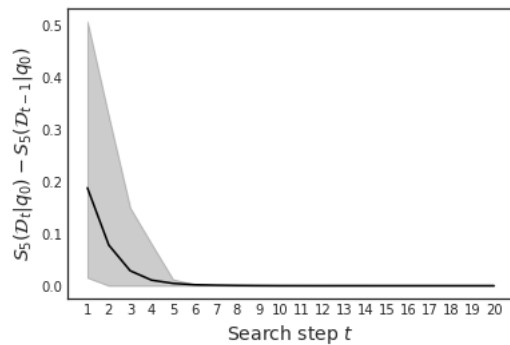

(a) Rocchio sessions' length

(b) Score gain at each search step

Figure 2: The histogram on the left shows the length of the Rocchio sessions, using different grammars on NQ Dev. The plot on the right shows the average score gain (score is computed according to Eq. 7) for each Rocchio expansion step with grammar G4 on NQ Dev. Shaded area is between $5 - 95^{th}$ percentiles.

find a first valid step, since Eq. 6a takes discrete values. Hence, we introduce a second component, $\text{NDCEM}_k$ (Eq. 6b) where $\text{em}(d|q) = 1$ if the answer extracted from $d$ by the reader exactly matches $a$, 0 otherwise. $\text{NDCEM}_k$ helps validate results by promoting high-ranking passages yielding correct answer spans. Finally, to favour high-confidence result sets we add the normalized Passage Score of the top $k$ results, leading to the following scoring function

$$\text{S}_k(\mathcal{D}|q) := (1 - \lambda_1 - \lambda_2) \cdot \text{NDCG}_k(\mathcal{D}|q) + \lambda_2 \cdot \text{NDCEM}_k(\mathcal{D}|q) + \lambda_1 \cdot \frac{1}{k} \sum_{i=1}^{k} \text{PS}(d_i|q) \quad \in [0, 1] \qquad (7)$$

Based on (7), we define the search step reward

$$r_t = \text{S}_5(\mathcal{D}_t|q_0) - \text{S}_5(\mathcal{D}_{t-1}|q_0). \qquad (8)$$

We train the MuZero agent directly on the reward. The reward is sparse, as none is issued in between search steps. The T5 agent is trained indirectly on the reward via the induction of Rocchio sessions (cf. §2.3).

## 5 Experiments

For our experiments we use the OpenQA-NQ dataset (Lee et al., 2019). This data is derived from Natural Questions (Kwiatkowski et al., 2019) and consists of Google queries paired with answers extracted from Wikipedia by human annotators. The data includes 79,168 train questions, 8,757 dev questions and 3,610 for test. We use the provided partitions and Wikipedia dump. Following Lee et al. (2019) we pre-process Wikipedia into blocks of 288 tokens, for a total of 13M passages. We evaluate each system on the top-5 288-token passages returned. Model selection and data analysis are performed on NQ Dev, using the reward (Eq. 8) as the objective.

### 5.1 Rocchio Sessions Data

We generate synthetic search sessions using Rocchio expansions for 5 different combinations of types of refinements. We refer to these as *grammars*: **G0** (allows only simple terms), **G1** (only term boosting, with weight $i \in \{0.1, 2, 4, 6, 8\}$), **G2** ('+' and '-'), **G3** (G0+G2) and **G4** (G0+G1+G2). Given the original query, a Rocchio session is generated as follows: We attempt at most $M = 100$ possible refinements for each grammar operator using terms from the constructed dictionaries $\Sigma_t^{\uparrow}$ and $\Sigma_t^{\downarrow}$ (see Eq. 3). For instance, for the '+' operator we attempt refinements of the form '+(*field*: "*term*")', where *term* is taken from the top-$M$ terms in the intersection dictionary $\Sigma_t^{\uparrow}$ and *field* represents the field (content or title) where such term was found. Dictionaries $\Sigma_t^{\uparrow}$ and $\Sigma_t^{\downarrow}$ are constructed (cf. §2.3) based on the set $\Sigma_t$ of top $N = 100$ terms present in the

documents retrieved so far, sorted according to Lucene's IDF score. For each of such possible refinements we issue the corresponding query to Lucene and, based on the returned documents, we evaluate the resulting score. We use the scoring function of Eq. 7 with coefficients $\lambda_1{=}0.2, \lambda_2{=}0.6$, after a search against the final quality metrics (cf. Appendix C). Then, we select the refinement leading to the highest score and neglect the other ones. This process continues until no score-improving refinement can be found, for a maximum of 20 refinement steps. A more formal description of the Rocchio session search procedure is summarized in Algorithm 1 in Appendix A, while examples of such sessions are reported in Table 1, Table A.9 and Table A.10.

In Figure 2a, we plot the histogram of the length of Rocchio sessions on NQ Dev, using the different grammars. We observe that most sessions terminate after a number of steps significantly smaller than 20, either because the maximum score is reached or because no score improvements can be found. For instance, using the G4 grammar, Rocchio sessions have an average length of 5.06 steps with standard deviation 3.28. Results are similar on NQ Train, where with grammar G4 we obtain 298,654 single Rocchio expansion steps from 77,492 questions (in Table A.1 we report the numbers for different grammars). Moreover, we have observed the first query expansion steps produce higher score gains with respect to later ones. This can be observed in Figure 2b where we plot the average per-step score's gain. This indicates that performing longer Rocchio expansions yields diminishing marginal gains.

## 5.2 Agents Training and Inference

The machine reader and passage scorer, as well as MuZero's $h_\theta$ function, use 12-layer BERT systems.[6] To train the former, we generate for each query in NQ Train 200 candidate passages from our BM25 system, picking one positive and 23 negative passages for each query at random whenever the query is encountered during training. The reader/scorer is not trained further. MuZero's representation function is trained jointly with the rest of the MuZero system.

For the T5 agent we start from the pretrained T5-11B (11 billion parameters) public checkpoint and continue training on the NQ Train Rocchio expansions. Training took about 5 days using 16 Cloud TPU v3. At inference time, we found that fixing the sessions to 20 steps worked best for both T5 and MuZero. Indeed, we observed performance increase monotonically with the search steps, with decreasing marginal gains (see Figure 4 where we plot the NQ Dev performance of one of our T5 agents as well as the supervised Rocchio sessions, as a function of the number of refinement steps). We report detailed training configurations and ablations in Appendix D.

The MuZero implementation is scaled and distributed via an agent-learner setup (Espeholt et al., 2018) in the SEED RL (Espeholt et al., 2020) framework allowing for centralized batching of inference for effective use of accelerators. MuZero is trained on NQ Train for a total of 1.6 million steps ($\approx$10 days) using 500 CPU-based actors and 4 Cloud TPU v2 for inference and training on the learner.[7] For each step, 100 simulations are performed. During training, we limit sessions to a maximum of 20 steps. The agent also can decide to stop early by selecting a dedicated stop action. Training of MuZero can be improved by providing *advice* to the actors. An actor may receive information about which terms $w_t$ should be promoted, $w_t \in \Sigma_t^\uparrow$, or demoted, $w_t \in \Sigma_t^\downarrow$. The probability of an episode receiving advice starts at 0.5 and decays linearly to 0 in one million steps.

## 5.3 Results

Table 2 summarizes the results on OpenQA-NQ Test. We evaluate passage retrieval quality by means of ranking (NDCG@5) and precision (Top-1, Top-5) metrics. We also report Exact Match (EM) to evaluate answer quality. The baseline is Lucene's BM25 one-shot search. Reranking the same BM25 documents by the PS score (BM25+PS) is easy and improves performance on all metrics, particularly noticeable on Top-1 and EM.[8] We also evaluate a pseudo relevance feedback variant of the BM25+PS baseline (+RM3). Following (Jaleel et al., 2004; Pal et al., 2013), at each iteration we pick the highest scoring term in the

---

[6]BERT-base, initialized from `https://tfhub.dev/google/bert_uncased_L-12_H-768_A-12/1`.

[7]For details, see https://cloud.google.com/tpu.

[8]Top-5 is identical to BM25 since the documents are the same.

Table 2: Results on the test partition of OpenQA-NQ. The **BM25** column reports the performance of the Lucene-BM25 search engine. **BM25+PS** refers to reranking the top-5 BM25 results with the BERT passage scorer (PS). **BM25+PS+RM3** is a pseudo-relevance feedback baseline that iteratively adds terms to the query and uses the passage scorer (PS) to aggregate the retrieved results. **MuZero** is the performance of the RL search agent using the full set of query expansion types (G4). **T5-G1** is the best T5 search agent, trained on the G1 grammar Rocchio sessions (using only term boosting). **MuZero+T5s** is an ensemble of the documents returned by the MuZero agent and all T5 agents, ranked based on each document PS score. For DPR's performance (**DPR**) we report the most recent Top-1 and Top-5 results from `https://github.com/facebookresearch/DPR`. Finally, **Rocchio-G4** is an estimate of the headroom based on the Rocchio sessions using the full grammar (G4). **NDCG@5**, **Top-1** and **Top-5** are retrieval quality metrics, while **EM** (Exact Match) is the answer quality metric used in machine reading.

| Metric | BM25 | +PS | +RM3 | MuZero | T5-G1 | MuZero+T5s | DPR | Rocchio-G4 |
|---|---|---|---|---|---|---|---|---|
| **NDCG@5** | 21.51 | 24.82 | 26.99 | 32.23 | 44.27 | 46.22 | - | 65.24 |
| **Top-1** | 28.67 | 44.93 | 46.13 | 47.97 | 52.60 | 54.29 | 52.47 | 73.74 |
| **Top-5** | 53.76 | 53.76 | 56.33 | 59.97 | 66.59 | 71.05 | 72.24 | 88.17 |
| **EM** | 28.53 | 41.14 | 40.14 | 32.60 | 44.04 | 44.35 | 41.50 | 62.35 |

search results based on the RM3 score, and add that term to the previous query with the '+' operator applied to the document content. In Appendix E.1 we provide a detailed study of the retrieval performance of this method, using all available operators, and comparing with an alternative IDF-based term selection mechanism. Surprisingly, and somewhat against the general intuition behind pseudo relevance feedback, we find that negating terms is more effective than promoting them. This seems to suggest that *negative* pseudo relevance feedback, in combination with reranking (e.g., by the PS score), can provide a simple and useful exploration device.

The last column (Rocchio-G4) reports the quality metrics for the best Rocchio sessions data, using the grammar with all operators (G4). Rocchio expansions make use of the gold answer and thus can be seen as a, possibly conservative, estimate of the performance upper bound. As the external benchmark we use DPR (Karpukhin et al., 2020), a popular neural retriever based on dual encoders, the dominant architecture for deep learning-based ad hoc retrieval (Craswell et al., 2020).

**T5**   We evaluate T5 models trained on all 5 grammar variants. The best one, 'T5-G1' in Table 2, is limited to term boosting (G1), and it learns to use all available weight values (Figure 3a). In terms of Top-1 this agent outperforms the published and the most recently posted DPR results[9] but has worse Top-5 than both. Results for all five T5 agents are found in Table A.6, we notice that the performance varies by relatively small amounts using different grammars, but it peaks noticeably with 'T5-G1'. Figure 4 shows the performance of the best Rocchio sessions data (Rocchio-G4) and that of the best T5 model (G1) as a function of the maximum number of steps allowed, both increasing monotonically as expected.

**MuZero**   On the retrieval task the MuZero agent outperforms all BM25 variants. While this result may seem trivial, it marked a milestone that required many iterations to achieve. The challenge for RL in IR, and NLU, is extreme in terms of state and action space dimensionality, data sparsity etc. (Zhang et al., 2021). Our ideas for tackling some of these key challenges by fitting out agents with domain knowledge in principled ways, with the grammar-guided MCTS as the centerpiece, seem to point in a promising direction. MuZero converges to a policy which uses only term boost action types with a weight of 2 – see Figure 3a for the action distributions of different policies. The MuZero agent is not able to find better-performing, diverse policies. This is an extreme case of a more general pattern. Different sub-grammars represent different tactics; e.g., '+' and '-' affect the accessible documents in irreversible ways, while boosting only affects ranking. It is challenging for all agents, and particularly MuZero, to modulate effectively multiple sub-policies.

---

[9]`https://github.com/facebookresearch/DPR`.

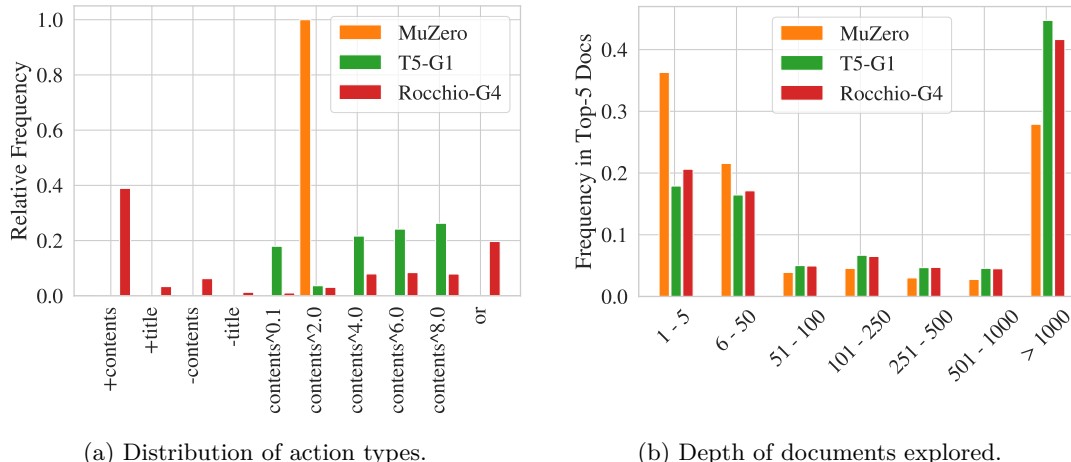

(a) Distribution of action types.

(b) Depth of documents explored.

Figure 3: The plot on the left shows the relative frequency of action types chosen by the best versions of the MuZero RL agent, the T5 agent that is learned on supervised episodes with the G1 grammar (only term boosting) 'T5-G1', and the Rocchio sessions with grammar G4 (complete grammar consisting of action types: simple terms, term boosting, '+', and '-') 'Rocchio-G4'. Interestingly, the MuZero agent converges to only use the 'light' boosting operation with a weight of 2. The T5 agent, on the other hand, makes use of the whole spectrum of the boosting operations, including the boosting with 0.1, which down-weights a particular term. The Rocchio query expansion uses the '+' operator on the contents field most often. This can be seen as an effective but potentially dangerous operation as it is a hard filtering on the presence of a certain term, potentially reducing the resulting retrieval set drastically. The right plot shows the depth of the documents in terms of retrieval rank based on the original query explored by the three agents. Here, we see that for all three agents, a significant portion of documents are retrieved beyond rank 1000, which means that they find relevant documents entirely hidden from a system relying on BM25 with only the original query.

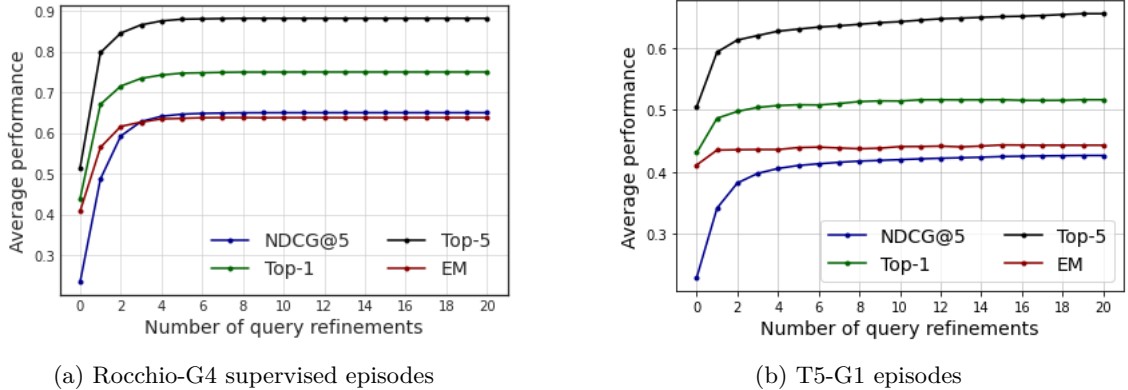

(a) Rocchio-G4 supervised episodes

(b) T5-G1 episodes

Figure 4: Performance on NQ Dev as a function of the number of query refinement steps. The plot on the left shows the results for the performance of the supervised Rocchio sessions with grammar G4 (all operators), while on the right we plot the performance of the trained T5-G1 agent trained on the G1 Rocchio sessions.

**Agents Ensemble** In the last experiment we combine all trained agents, the five T5 agents and MuZero, in one ensemble. We simply rank the union of all the documents returned by the ensemble, by means of the PS score on each document, thus not requiring any additional parameters. This ensemble ('MuZero+T5s' in Table 2) has slightly better precision than the recent DPR in top position, and slightly worse for the Top-5. This results indicates that the ability to orchestrate diverse sub-policies may indeed be the key to future progress for search agents. For the record, the current SOTA for Top-5 is 74.0 (Qu et al., 2021).

Table 3: Example of a T5-G4 agent session exhibiting multiple tactics. The session shows the evolution of the search query (first line in each section) and snippets of the top-3 retrieved documents from the search engine. We skip $q_1$ and $q_2$ for brevity. The colored spans indicate the prediction of the machine reader; blue if it is correctly predicted, red otherwise. In the top right corner of each section, we report the score of the retrieved document set at that step, according to Equation 7.

| | Query and Search Results | Score |
|---|---|---|
| $q_0$ | who averaged the most points in college basketball history | |
| | Top-3 documents retrieved with $q_0$: | 0.027 |
| $d_1$ | Title    Gary Hill (basketball) | |
| | Content  . . . one of four on that team who averaged double figures in points. Senior **Larry Jones** was OCU's leading scorer at 19.7 points a game, sophomore Bud Koper added 15.9. . . | |
| $d_2$ | Title    Kevin Foster (basketball) | |
| | Content  . . . his senior year, Foster averaged 21 points per game and was named the MVP and All-District 18-5A First Team. He was also a Texas top- **30** player his final season . . . | |
| $d_3$ | Title    Paul Cummins (basketball) | |
| | Content  . . . big home win over Army. As a freshman, Cummins high-scored with **13** points against final-four team Louisville (2004). After graduating in 2008, Cummins played for . . . | |
| $q_3$ | who averaged the most points in college basketball history (contents:"per"∧6) (contents:"scorer"∧4) (contents:"3"∧6) | |
| | Top-3 documents retrieved with $q_3$: | 0.330 |
| $d_1$ | Title    Alphonso Ford | |
| | Content  . . . seasons. With 3,165 career points scored in the NCAA Division I, he is 4th on the all-time scoring list, behind only **Pete Maravich**, Freeman Williams, and Lionel . . . | |
| $d_2$ | Title    Buzzy Wilkinson | |
| | Content  **Buzzy Wilkinson** Richard Warren "Buzzy" Wilkinson (November 18, 1932 – January 15, 2016) was an American basketball player who was selected by the Boston Celtics in . . . | |
| $d_3$ | Title    Gary Hill (basketball) | |
| | Content  . . . becoming one of four on that team who averaged double figures in points. Senior **Larry Jones** was OCU's leading scorer at 19.7 points a game, sophomore Bud Koper . . . | |
| $q_4$ | who averaged the most points in college basketball history (contents:"per"∧6) (contents:"scorer"∧4) (contents:"3"∧6) +(contents:"maravich") | |
| | Top-3 documents retrieved with $q_4$: | 0.784 |
| $d_1$ | Title    Alphonso Ford | |
| | Content  . . . seasons. With 3,165 career points scored in the NCAA Division I, he is 4th on the all-time scoring list, behind only **Pete Maravich**, Freeman Williams, and Lionel . . . | |
| $d_2$ | Title    Pete Maravich | |
| | Content  . . . had posted a 3–20 record in the season prior to his arrival. **Pete Maravich** finished his college career in the 1970 National Invitation Tournament, where LSU finished fourth . . . | |
| $d_3$ | Title    1970 National Invitation Tournament | |
| | Content  . . . represented the final college games for LSU great **Pete Maravich**, the NCAA's all-time leading scorer. Maravich finished his three-year career with 3,667 points . . . | |

**Answer Quality**   We conclude by discussing answer quality. Agents routinely produce answer spans, as predicted by the reader/scorer, to build observations. The MR/PS component is trained once, before the agents, on the output of BM25. However, agents deeply affect the results composition. As Figure 3b shows, search agents dig deep in the original BM25 ranking. This is positive, as behavior discovery is one of the main motivations for researching exploratory methods like RL. As a consequence, though, the MR/PS component effectively operates out of distribution and the EM numbers of the internal reader are not competitive with

recent methods, Table A.7 reports all the numbers including on NQ Dev. Ideally, one would co-train the observation builder with the search agent. However, combining the two would introduce significant engineering complexity in the current architecture. For instance, one could interleave training the two as in DQNs (Mnih et al., 2013).

A simpler alternative is to add the answer prediction task to the T5 agent. Retrieval-augmented answer generation is known to produce strong results (Izacard & Grave, 2021). Multitasking would simplify the design of the generative agents and possibly produce better models. We make a first step in this direction by training a dedicated T5 agent. The system uses as training input the top-5 documents of the Rocchio-G4 episodes, but its task is to generate the gold answer, instead of the query expansion. At evaluation time, based on the output of the 'T5-G1' and 'MZ+T5s' agents, the EM performance of the answer generation T5 is comparable to methods that build on DPR, such as RAG (Lewis et al., 2020b) (44.5 EM). Although not as good as FID (Izacard & Grave, 2021) that condition on many more (100) documents.

### 5.4 Discussion

**Limitations of Current Policies**   Table 3 illustrates an example where the T5-G4 agent (with the full set of operators) switches policy mid-session. The question is about basketball records and BM25 does not find good results. In the first three steps the agent focuses on *re-ranking* by boosting terms like 'per' (from the phrase 'per game' in the results for $q_0$) and 'scorer'. This produces a good hit and predicted answer span ('Pete Maravich') at position 1 of step 3. The agent then switches to *filtering* mode, to focus on documents containing the answer term predicted by the machine reader. While this is a clear instance of successful policy synthesis, the T5-G4 agent does not master switching between policies well enough to perform better than T5-G1, the agent that only uses boost operators. Table 4 provides an example that shows how T5-G1 is more robust than T5-G4. T5-G4 starts by requiring the presence of a misspelled term ('highschool') which leads to empty results and the end of the session because that step is not reversible. T5-G1, instead, makes its way gradually in the session boosting topical terms ('draftees') and players names eventually solving the query.

The agents ensemble results prove that the ability to orchestrate complementary sub-policies provides a performance advantage. This suggests that the action space may benefit by including more control actions, e.g. to 'undo' or 'go back' to a specific state, to better support safe exploration and the emergence of meta policies. We plan to investigate this in future work. The previous point extends to the agents' architecture. It is reasonable to hypothesise that the superior performance of T5 is due to two main factors. T5s are bigger models, trained on more data, and rely on a more powerful prediction process based on the encoder-decoder architecture. In addition, they are finetuned on a self-supervised task which provides significant headroom. While large LMs seem the obvious choice forward there are open questions concerning exploration. It is not clear how much the model can generalize, being trained offline and never being exposed to its own predictions. This moves the learning problem back towards RL. We have started to investigate approaches in the direction of decision/trajectory transformers (Chen et al., 2021; Janner et al., 2021). We believe they provide a natural framework for bringing back key RL concepts which could play an important role; for example, by allowing successful policy synthesis by training from different offline policies; e.g., from Rocchio and MuZero.

**Artificial vs Human Search Policies**   Based on human search behavior (cf. §2), it seems natural to model search as an iterative, contextualized machine learning process. In terms of the number of steps required, Rocchio sessions peak at around 5 steps, while also for humans, especially for hard queries, several step are often necessary. Qualitatively speaking, though, they look different. For a start, while powerful, search operators (at least in the current form) don't allow to easily capture the full spectrum of human search tactics. Human search sessions have been characterized broadly in terms of three types of refinement actions: specification, generalization and reformulation (Lau & Horvitz, 1999; Downey et al., 2008). In this respect the current current search agents lack the ability to explicitly generalize and fully reformulate. They mostly perform filtering and reranking. Search operators may be better suited to complement, as power tools, other plain language query refinement methods rather than being the centerpiece of the agent's action space. Evaluating plain language reformulation functionality is thus an obvious next step. However, the generation of the necessary training data in this case is an open question. We will focus on this problem in future work.

We also point out that the policies that can be currently generated via the Rocchio sessions, or by exploration via Muzero, are artificial because they are driven by a reward which is an imperfect proxy for human relevance. In future work, we plan to investigate new learning methods that include modeling of human policies, e.g., in combination with apprenticeship learning frameworks (cf. (Nakano et al., 2021)).

**Thoughts on OpenQA-NQ**  The Natural Questions dataset (Kwiatkowski et al., 2019) is unique in that it builds from real user queries, with a great deal of attention to annotation and data quality. On the other hand, the dataset is designed for a setup where the document is given. Hence, annotations are consistent only within that document, not at the collection level. The retrieval setting implies that the vast majority of the data have not been validated by raters. Additionally, the human ratings cannot be easily and reliably aligned with a pre-computed segmentation into passages. Thus, one typically relies on the heuristic relevance function, based on the presence of the short answer string, which cannot discriminate unjustified answers. While imperfect, this setup strikes a local optimum that has driven significant innovation in IR and QA research by allowing direct comparison of many different approaches in a fast moving landscape; e.g. from ORQA (Lee et al., 2019) to closed book QA (Roberts et al., 2020) to RAG (Lewis et al., 2020b; Qu et al., 2021), DPR (Karpukhin et al., 2020) etc. Another possible downside is the overlap between partitions, as pointed out in (Lewis et al., 2021a). We controlled for this factor periodically by splitting the dev partition into known and unknown answers (based on the presence of the answer in the train data). Consistently with (Lewis et al., 2021a) we find a significant drop on the unknown answers but the same relative performance of methods.

**Broader Impact**  We would like to note that pre-trained language models of the kind used here have been shown to capture societal biases (Tan & Celis, 2019; Webster et al., 2020), which motivates a broad discussion about potential harms and mitigations (Blodgett et al., 2020; Bender et al., 2021). We have no reason to believe our architectures would exacerbate biases, but the overall problems may persist. We also hope that end-to-end optimization methods based on composite rewards, as in this proposal, can contribute to addressing some of these challenges; e.g., by providing means of adversarial testing, and by including relevant metrics directly in the objective design. We stress here that, while our agents yield performance comparable to neural retrievers, they rely solely on interpretable, transparent, symbolic retrieval operations.

# 6 Related Work

Query optimization is an established topic in IR. Methods range from hand-crafted rules (Lawrence & Giles, 1998) to data-driven transformation patterns (Agichtein et al., 2001). Narasimhan et al. (2016) use RL to query the web for information extraction. Nogueira & Cho (2017) and Buck et al. (2018) use RL-trained agents to seek good answers by reformulating questions with seq2seq models. These methods are limited to one-step episodes and queries to plain natural language. This type of modeling is closely related to the use of RL for neural machine translation, whose robustness is currently debated (Choshen et al., 2020; Kiegeland & Kreutzer, 2021). Montazeralghaem et al. (2020) propose a feature-based network to score potential relevance feedback terms to expand a query. Das et al. (2019) propose to perform query reformulation in embedding (continuous) space and find that it can outperform the sequence-based approach. Xiong et al. (2021) successfully use relevance feedback by jointly encoding the question and the text of its retrieved results for multi-hop QA. Other work at the intersection of IR and RL concerns bandit methods for news recommendation (Li et al., 2010) and learning to rank (Yue & Joachims, 2009). Recently, interest in Deep RL for IR has grown (Zhang et al., 2021). There, the search engine is the agent, and the user the environment. In contrast, we view the search problem from the user perspective and thus consider the search engine as the environment.

The literature on searchers' behavior is vast, see e.g. Strzelecki (2020) for an overview of eye-tracking studies. While behavior evolves with interfaces, users keep parsing results fast and frugally, attending to just a few items. From a similar angle, Yuan et al. (2020) offer promising findings on training QA agents with RL for template-based information gathering and answering actions. Most of the work in language-related RL is otherwise centered on synthetic navigation/arcade environments (Hu et al., 2019). This line of research shows that RL for text reading can help transfer (Narasimhan et al., 2018) and generalization (Zhong et al.,

2020) in synthetic tasks but skirts the challenges of more realistic language-based problems. On the topic of grammars, Neu & Szepesvári (2009) show that Inverse RL can learn parsing algorithms in combination with PCFGs (Salomaa, 1969).

Current work in OpenQA focuses on the search engine side of the task, typically using dense neural passage retrievers based on a dual encoder framework instead of BM25 (Lee et al., 2019; Karpukhin et al., 2020). Leveraging large pre-trained language models to encode the query and the paragraphs separately led to a performance boost across multiple datasets, not just in the retrieval metrics but also in exact-match score. While Karpukhin et al. (2020) use an extractive reader on the top-k returned paragraphs, Lewis et al. (2020b) further improves using a generative reader (BART (Lewis et al., 2020a)). This design combines the strengths of a parametric memory – the pre-trained LM – with a non-parametric memory – the retrieved Wikipedia passages supplied into the reader's context. This idea of combining a dense retriever with a generative reader is further refined in Izacard & Grave (2021), which fuses multiple documents in the decoding step. A recent line of work is concerned with constraining the model in terms of the number of parameters or retrieval corpus size while remaining close to state-of-the-art performance (Min et al., 2021). This effort led to a synthetic dataset of 65 million *probably asked* questions (Lewis et al., 2021b) used to do a nearest neighbor search on the question – no learned parameters needed – or train a closed-book generative model.

## 7   Conclusion

Learning to search sets an aspiring goal for AI, touching on key challenges in NLU and ML, with far reaching consequences for making the world's knowledge more accessible. Our paper provides the following contributions. First, we open up the area of search session research to supervised language modeling. Second, we provide evidence for the ability of RL to discover successful search policies in a task characterized by multi-step episodes, sparse rewards and a high-dimensional, compositional action space. Lastly, we show how the search process can be modeled via transparent, interpretable machine actions that build on principled and well-established results in IR and NLU.

Our findings seem to agree with a long-standing tradition in psychology that argues against radical behaviorism – i.e., pure reinforcement-driven learning, from *tabula rasa* – for language (Chomsky, 1959). RL agents require a remarkable share of hard-wired domain knowledge. LM-based agents are easier to put to use, because they rely on massive pre-training and abundant task-specific data for fine tuning. Supplied with the right inductive bias, LM and RL search agents prove surprisingly effective. Different architectures learn different, complementary, policies, suggesting broad possibilities in the design space for future work.

## Acknowledgments

We would like to thank for their feedback: Robert Baldock, Marc Bellemare, Jannis Bulian, Michelangelo Diligenti, Sylvain Gelly, Thomas Hubert, Rudolf Kadlec, Kenton Lee, Simon Schmitt, Julian Schrittwieser, David Silver. We also thank the reviewers and action editor for their valuable comments and suggestions.

Table 4: Snippet of episode examples from the T5-G1 (boosting only) agent vs. the T5-G4 agent (all operators). The best performing T5 agent makes use of the boosting-only grammar. This example showcases one reason that might explain the superiority of this particular grammar. The BM25 results for the initial query, do not lead to satisfactory results, with a score of 0.081. The T5-G1 adjustments to the query; first, boosting "draftees", and later boosting "thon", and "satnam" leads to almost perfect retrieval results with a score of 0.946. On the other hand, the T5-G4 agent decides to constraint the results in the first step to those including the term "highschool". While this is a topical term, this leads to a bad retrieval results set from which the agent cannot recover in later steps (omitted for brevity). The reason for this becomes apparent when inspecting the good search results of the T5-G1 agent: they do not contain the term "highschool", but the terms "high school" or "high schoolers". The constraint action ("+") filters these good documents out.

| | | Query and Search Results | Score |
|---|---|---|---|
| $q_0$ | | who was the last nba player to get drafted out of highschool | |
| Top-2 documents retrieved with $q_0$: | | | 0.081 |
| $d_1$ | Title | 1996 NBA draft | |
| | Content | . . . Jermaine O'Neal, Peja Stojaković, Antoine Walker), and one undrafted All-Star (**Ben Wallace**), for a grand total of 11 All-Stars. . . . | |
| $d_2$ | Title | 2009 NBA draft | |
| | Content | . . . The **2009** draft marked the first time three sons of former NBA players were selected in the top 15 picks of the draft . . . | |
| **T5-G1 $q_1$** | | who was the last nba player to get drafted out of highschool (contents:"draftees"∧2) | |
| Top-2 documents after aggregation with retrieval results from **T5-G1 $q_1$**: | | | 0.374 |
| $d_1$ | Title | NBA high school draftees | |
| | Content | . . . hold themselves back a year in **high school** before declaring for the draft, like with Satnam Singh Bhamara or **Thon Maker**. The NBA has long had a preference for players who played basketball at the collegiate level . . . | |
| $d_1$ | Title | 1996 NBA draft | |
| | Content | . . . Jermaine O'Neal, Peja Stojaković, Antoine Walker), and one undrafted All-Star (**Ben Wallace**), for a grand total of 11 All-Stars. . . . | |
| **T5-G1 $q_3$** | | who was the last nba player to get drafted out of highschool (contents:"draftees"∧2) (contents:"thon"∧4) (contents:"satnam"∧4) | |
| Top-2 documents after aggregation with retrieval results from **T5-G1 $q_3$**: | | | 0.946 |
| $d_1$ | Title | NBA draft | |
| | Content | . . . However, because of the new age requirement put in place in 2005, **high school** seniors are no longer eligible for the draft, unless they were declared as postgraduates by the NBA, which would not happen until 2015 with Indian prospect Satnam Singh Bhamara in the second round and again in 2016 with South Sudanese–Australian prospect **Thon Maker** in the first round. . . . | |
| $d_2$ | Title | Eligibility for the NBA draft | |
| | Content | . . . However, in recent years, other players like Satnam Singh, **Thon Maker**, and Matur Maker have looked to enter the NBA draft while still being **high schoolers** by exploiting a loophole where they enter the draft as **high school** postgraduates. . . . | |
| **T5-G4 $q_1$** | | who was the last nba player to get drafted out of highschool +(contents:"highschool") | |
| Top-2 documents after aggregation with retrieval results from **T5-G4 $q_1$**: | | | 0.081 |
| $d_1$ | Title | 1996 NBA draft | |
| | Content | . . . Jermaine O'Neal, Peja Stojaković, Antoine Walker), and one undrafted All-Star (**Ben Wallace**), for a grand total of 11 All-Stars. . . . | |
| $d_2$ | Title | 2009 NBA draft | |
| | Content | . . . The **2009** draft marked the first time three sons of former NBA players were selected in the top 15 picks of the draft . . . | |

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

# Appendix

## A  Rocchio Sessions

---

**Algorithm 1:** Rocchio Sessions

---

**input**  : A question-answer pair $(q, a)$, $\mathtt{k} = 5$, $\mathtt{num\_steps} = 20, \mathtt{N} = 100, \mathtt{M} = 100$
**output**: A set of observation-query expansion pairs for training a T5 agent $\mathtt{RQE} = \{(o_t, \Delta q_t)\}$

$\mathtt{RQE} \leftarrow \emptyset$  $q_t \leftarrow q$;
$\mathcal{D}_t \leftarrow \emptyset$ ;                                                    // Unique documents found in the session
$q_* \leftarrow q +(\text{contents:``a''})$ ;                                               // The ideal query
$\mathcal{D}_* \leftarrow \mathtt{LuceneBM25}(q_*)$ ;                                    // Use search to get the top $k$ documents
// Use the agent PS and MR components to rerank the documents, extract answer spans to compute
   the snippets from the top $k$ results, and compile the observation (cf.  also Appendix B)
$o_* \leftarrow \mathtt{ComputeObservation}(q, q_*, \mathcal{D}_*, k)$;
$\Sigma_* \leftarrow \mathtt{TopNTermsByLuceneIDF}(o_*, \mathtt{N})$ ;                        // Collect good search terms
**for** $t \leftarrow 1$ **to** num_steps **do**
 $\mathcal{D}_t \leftarrow \mathcal{D}_t \cup \mathtt{LuceneBM25}(q_t)$;
 $o_t \leftarrow \mathtt{ComputeObservation}(q, q_t, \mathcal{D}_t, k)$;
 $\Sigma_t \leftarrow \mathtt{TopNTermsByLuceneIDF}(o_t, \mathtt{N})$;
 $\Sigma_t^{\uparrow} \leftarrow \Sigma_* \cap \Sigma_t, \Sigma_t^{\downarrow} \leftarrow \Sigma_t - \Sigma_*$;
 $s_t \leftarrow \mathtt{ComputeScore}(q, \mathcal{D}_t, k)$;                            // Compute the score using Eq.(7)
 $\mathtt{max\_score} \leftarrow s_t$;
 $\mathtt{best\_action} \leftarrow \emptyset$;
 // Evaluate all available operators
 **for** op $\in \{+, -, \wedge 0.1, \wedge 2, \wedge 4, \wedge 6, \wedge 8, `\ '\}$ **do**
  $\mathtt{num\_tries} \leftarrow 0$;
  **for** $w \ \in \Sigma_t^{\uparrow} \cup \Sigma_t^{\downarrow}$ **do**
   **if** (op $=='-' \wedge w \in \Sigma^{\downarrow}$) $\vee$ (op $\neq '-' \wedge w \in \Sigma^{\uparrow}$) $\wedge$ (num_tries $<$ M) **then**
    $\Delta q_t \leftarrow \text{op}(w.\mathtt{field}, w.\mathtt{term})$;          // Query refinement according to semantic operator
    $q' \leftarrow q_t + \Delta q_t$;
    $\mathcal{D}' \leftarrow \mathcal{D}_t \cup \mathtt{LuceneBM25}(q')$;
    $s' \leftarrow \mathtt{ComputeScore}(q, \mathcal{D}', k)$;
    $\mathtt{num\_tries} \leftarrow \mathtt{num\_tries} + 1$;
    **if** $s' > \mathtt{max\_score}$ **then**
     $\mathtt{max\_score} \leftarrow s'$;
     $\mathtt{best\_action} \leftarrow \Delta q_t$;
    **end**
   **else**
    continue;
   **end**
  **end**
 **end**
 **if** max_score $> s_t$ **then**
  // If the best action improves the score, add this step to the data, and continue the
      session
  $q_t \leftarrow q_t + \mathtt{best\_action}$;
  $\mathtt{RQE} \leftarrow \mathtt{RQE} \cup (o_t, \mathtt{best\_action})$;
 **else**
  **return** $\mathtt{RQE}$;
 **end**
**end**
**return** $\mathtt{RQE}$

---

Algorithm 1 provides a schematic summary of the procedure for generating Rocchio sessions described in §5.1, using the full set of grammar operators (G4). We omit the terms source for simplicity and readability, but it should be straightforward to reconstruct. Table A.10 shows another example of such a Rocchio expansion session.

In Table A.1 below, we report the total number of expansion steps performed on NQ Train. These are used as supervised training data for our T5 agents.

| **G0** | **G1** | **G2** | **G3** | **G4** |
|--------|--------|--------|--------|--------|
| 243,529 | 313,554 | 230,921 | 246,704 | 298,654 |

Table A.1: Total number of Rocchio expansion steps in NQ Train for different grammars on the 77,492 Rocchio sessions.

## B   Observation Building Details

This section provides more details and examples about the encoding of observations for both the MuZero and the T5 agent. As described in Section 2.2, the main part of the observation consists of the top-5 documents from all results retrieved so far, $\cup_{i=0}^{t} \mathcal{D}_i$. The documents are sorted according to the PS score and reduced in size by extracting fixed-length snippets around the machine reader's predicted answer. Moreover, the corresponding Wikipedia article title is appended to each document snippet. The computational complexity of this step is determined by running a BERT-base (110M parameters) machine reader separately (albeit possibly in parallel) over five passages. In addition to the top documents, the observation includes the original question and information about any previous refinements. While the main part of the observation is shared between the MuZero and the T5 agent, there are differences in the exact representation. The following two paragraphs give a detailed explanation and example for both agents.

### B.1   MuZero Agent's State (cf. §2.2)

The MuZero agent uses a custom BERT (initialized from BERT-base) with additional embedding layers to represent the different parts of the observation. It consists of four individual embedding layers as depicted in Figure A.1. At first, the standard layer for the tokens of the query, the current tree, and the current top-5 documents $D$. The second layer assigns a type ID to each of the tokens representing if a token is part of the query, the tree, the predicted answer, the context, or the title of a document. The last two layers add scoring information about the tokens as float values. We encode both the inverse document frequency (IDF) of a word and the documents' passage selection (PS) score. Figure A.2 shows a concrete example of a state used by the MuZero agent.

| **Layer** | **Query** | **Tree** | **Document Results** |
|-----------|-----------|----------|----------------------|
| Tokens | $q_0$ | $l_0, \ldots, l_m$ | $a_0, c_0, t_0, \ldots, a_n, c_n, t_n$ |
| Type | $\mathrm{ID}_q$ | $\mathrm{ID}_{\mathrm{tree}}$ | $\mathrm{ID}_a, \mathrm{ID}_c, \mathrm{ID}_t, \ldots, \mathrm{ID}_a, \mathrm{ID}_c, \mathrm{ID}_t$ |
| IDF Score | $\mathrm{idf}(q_0)$ | $\mathrm{idf}(l_0), \ldots, \mathrm{idf}(l_m)$ | $\mathrm{idf}(a_0), \mathrm{idf}(c_0), \mathrm{idf}(t_0), \ldots, \mathrm{idf}(a_n), \mathrm{idf}(c_n), \mathrm{idf}(t_n)$ |
| PS Score | $0$ | $0$ | $\mathrm{PS}(d_0), \ldots, \mathrm{PS}(d_n)$ |

Figure A.1: Schematic illustration of the MuZero search agent's state for the BERT representation function.

Table A.2: Example state of the MuZero search agent that is the input to the BERT representation function. The 'Type' layer encodes the state part information for each token. The 'IDF' and 'PS' layer are additional layers with float values of the IDF and the PS score of the input tokens, respectively.

| Tokens | [CLS] | who | carries | the | burden | of | going | forward | with | evidence | in | a | trial | |
|---|---|---|---|---|---|---|---|---|---|---|---|---|---|---|
| **Type** | [CLS] | query | query | query | query | query | query | query | query | query | query | query | query | |
| **IDF** | 0.00 | 0.00 | 6.77 | 0.00 | 7.77 | 0.00 | 5.13 | 5.53 | 0.00 | 5.28 | 0.00 | 0.00 | 5.77 | ... |
| **PS** | 0.00 | 0.00 | 0.00 | 0.00 | 0.00 | 0.00 | 0.00 | 0.00 | 0.00 | 0.00 | 0.00 | 0.00 | 0.00 | |

| Tokens | [SEP] | [pos] | [content] | burden | ##s | [neg] | [title] | sometimes | [SEP] | lit | ##igan | ##ts | [SEP] | |
|---|---|---|---|---|---|---|---|---|---|---|---|---|---|---|
| **Type** | [SEP] | tree | tree | tree | tree | tree | tree | tree | [SEP] | answer | answer | answer | [SEP] | |
| **IDF** | 0.00 | 0.00 | 0.00 | 9.64 | 9.64 | 0.00 | 0.00 | 4.92 | 0.00 | 10.64 | 10.64 | 10.64 | 0.00 | ... |
| **PS** | 0.00 | 0.00 | 0.00 | 0.00 | 0.00 | 0.00 | 0.00 | 0.00 | 0.00 | -3.80 | -3.80 | -3.80 | -3.80 | |

| Tokens | kinds | for | each | party | , | in | different | phases | of | litigation | . | the | burden | |
|---|---|---|---|---|---|---|---|---|---|---|---|---|---|---|
| **Type** | context | context | context | context | context | context | context | context | context | context | context | context | context | |
| **IDF** | 7.10 | 0.00 | 0.00 | 4.36 | 17.41 | 0.00 | 4.18 | 7.46 | 0.00 | 7.92 | 17.41 | 0.00 | 7.77 | ... |
| **PS** | -3.80 | -3.80 | -3.80 | -3.80 | -3.80 | -3.80 | -3.80 | -3.80 | -3.80 | -3.80 | -3.80 | -3.80 | -3.80 | |

| Tokens | suspicion | " | , | " | probable | cause | " | ( | as | for | [SEP] | evidence | [SEP] | |
|---|---|---|---|---|---|---|---|---|---|---|---|---|---|---|
| **Type** | context | context | context | context | context | context | context | context | context | context | [SEP] | title | [SEP] | |
| **IDF** | 7.80 | 17.41 | 17.41 | 17.41 | 7.91 | 5.41 | 17.41 | 17.41 | 0.00 | 0.00 | 0.00 | 5.28 | 0.00 | ... |
| **PS** | -12.20 | -12.20 | -12.20 | -12.20 | -12.20 | -12.20 | -12.20 | -12.20 | -12.20 | -12.20 | -12.20 | -12.20 | -12.20 | |

## B.2 T5 Agent's State (cf. §3.1)

T5 represents the state as a flat string. The input is a concatenation of the original query, zero or more expansions, and five results. For each result, we include the answer given by the reader, the document's title, and a span centered around the answer. The prediction target is simply the next expansion. See Table A.3 for a full example.

Table A.3: Example state (input) and prediction (target) of the T5 agent with linebreaks and emphasis added for readability. We use a 30 token span in our experiments.

| | |
|---|---|
| Input | Query: 'how many parts does chronicles of narnia have'. 
 Contents must contain: lewis. 
 Contents cannot contain: battle boost 2.0. 

 Answer: 'seven'. 
 Title: 'The Chronicles of Narnia'. 
 Result: The Chronicles of Narnia is a series of *seven* fantasy novels by C. S. Lewis. It is considered a classic of children's literature and is the author's best-known work, having... 

 Answer: 'seven'. 
 Title: 'The Chronicles of Narnia (film series)'. 
 Result: '"The Chronicles of Narnia", a series of novels by C. S. Lewis. From the *seven* books, there have been three film adaptations so far – (2005), "" (2008) and "" (2010)... 

 Answer: 'seven'. 
 Title: 'Religion in The Chronicles of Narnia'. 
 Result: 'Religion in The Chronicles of Narnia "The Chronicles of Narnia" is a series of *seven* fantasy novels for children written by C. S. Lewis. It is considered a classic of... 

 Answer: 'seven'. 
 Title: 'The Chronicles of Narnia'. 
 Result: 'Lewis's early life has parallels with "The Chronicles of Narnia". At the age of *seven* , he moved with his family to a large house on the edge of Belfast... 

 Answer: 'Two'. 
 Title: 'The Chronicles of Narnia'. 
 Result: 'found in the most recent HarperCollins 2006 hardcover edition of "The Chronicles of Narnia". *Two* other maps were produced as a result of the popularity of the 2005 film ... |
| Target | Contents must contain: novels |

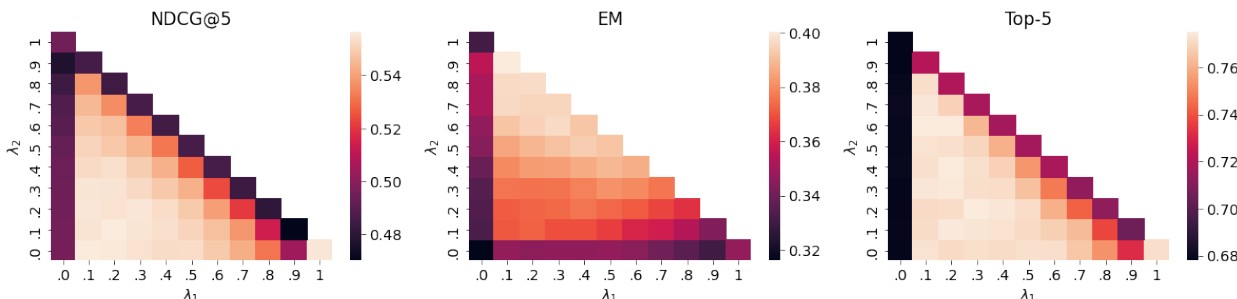

Figure A.2: Performance (NDCG@5, EM, and Top-5, respectively) of the Rocchio episodes from NQ-dev guided by the composite score of Equation 7, as a function of the coefficients $\lambda_1$ and $\lambda_2$.

## C Reward details

We investigate the effects of the three score components in the definition of the composite scoring function of Eq. 7. As mentioned in Section 4, in our experiments we have observed that using only the $\text{NDCG}_k$ score as reward signal (i.e., setting $\lambda_1 = \lambda_2 = 0$ in Eq. 7) has several limitations. This motivated us to introduce the $\text{NDCEM}_k$ and PS components with the intent of: 1) providing further guidance to the agent (whenever $\text{NDCG}_k$ cannot be increased, the agent can further refine the query by increasing $\text{NDCEM}_k$ or PS), and 2) regularizing the search episodes by making the score more robust with respect to exploratory behaviors that could lead to drift.

We run a grid search over the reward coefficients $\lambda_1, \lambda_2$ and, for each of their values, we evaluate the performance of the Rocchio sessions on NQ Dev (for a high throughput, we select grammar **G3** and set $N = M = 20$). Figure A.2 shows the respective end-to-end performance in terms of our three main quality metrics: NDCG@5, EM, and Top-5.

The results in Figure A.2 support our intents: by introducing $\text{NDCEM}_k$ and PS scores in the reward (i.e., setting $\lambda_1, \lambda_2 > 0$), the Rocchio expansions can achieve significantly higher performance, in all the three metrics, with respect to using only an $\text{NDCG}_k$ score ($\lambda_1 = \lambda_2 = 0$) (notably, it improves also the NDCG@5 itself, meaning that the agent is not trading-off performance metrics but it is indeed producing higher quality sessions). It is also worth pointing out the role of the $\text{NDCEM}_k$ score component, weighted by coefficient $\lambda_2$. Notice that good NDCG@5 and Top-5 performance could be achieved also setting $\lambda_2 = 0$ (see, e.g., the bottom-right corner $\lambda_1 = 1, \lambda_2 = 0$). However, this leads to definitely worse EM results compared to when $\lambda_2 > 0$. Intuitively, a $\text{NDCEM}_k$ component $\lambda_2 > 0$ ensures that the returned documents, in addition to containing the gold answer (thus having high NDCG@5 and Top-5), are also relevant for the query (thus reaching a high EM). Hence, it is crucial to prevent semantic drifts. Based on these results we set $\lambda_1 = 0.2, \lambda_2 = 0.6$, which is a sweet spot in Figure A.2.

## D Model, Training Configuration, and Computational Complexity

### D.1 MuZero

The MuZero agent learner, which performs both inference and training, runs on a Cloud TPU v2 with 8 cores which is roughly equivalent to 10 Nvidia P100 GPUs in terms of TFLOPS.[10] One core is allocated for training and 7 cores are allocated for inference. We use 500 CPU based actors along with 80 actors dedicated to evaluation. Each agent is trained for 1.6 million steps, with 100 simulations per step, at an approximate speed of 10,000 steps per hour. In total, training takes about 10 days. Hyperparameters are listed in Table A.4.

---

[10]The Cloud TPU v2 has a peak performance of 180 TFLOPS (https://cloud.google.com/tpu), whereas the Nvidia P100 GPU goes up to 18.7 TFLOPS depending on precision (https://cloud.google.com/compute/docs/gpus).

**Computational Complexity**  The computational complexity of a single step of the MuZero agent is determined by the complexity of the state encoding function ("h" in Figure A.3b) and the number of simulations during MCTS. For the state encoding function, we use BERT-base to encode the state and a GRU cell with a hidden size of 32 to encode the past actions. The maximum sequence length of the state is 512. The recurrent inference function during the MCTS ("g" in Figure A.3b) is an LSTM with a hidden dimension of 512 that is invoked for each of the number of simulations (typically 100). On top of the LSTM representation, we use MLPs ("f" in Figure A.3b) with a single hidden layer with hidden dimension 512 as the policy, value, and reward head.

**Tuning**  We carried out extensive model selection and tuning while implementing the MuZero algorithm using TicTacToe and Atari environments, and based on the information available in the original paper (Schrittwieser et al., 2020). We ran these experiments on a single TPU, to ensure the replicability of healthy training runs without extensive computing resources. We did not try to match the performance of the MuZero paper on the Atari and board games, because that would have required significantly more compute resources and time and it was beyond the scope of our project. Optimizing MuZero can be hard because of the many hyperparameters, especially in the learning to search task where training can take a long time to stabilize. Thus, we relied to a large extent on the existing configuration and attempted primarily to optimize the MCTS process. We tried to find better parameters – via simple grid search – of prioritized and importance sampling (exponents), replay buffer and queue (sizes), action selection softmax (temperature), exploration noise (Dirichlet $\alpha$), the $c_1$, $c_2$ parameters of the Upper Confidence Bound score and the number of simulations. In the end the main parameter that consistently affected performance was the number of simulations. More simulations yield better performance, and slower training. We observed diminishing improvements after 100 simulations and settled on that as the final value. We also experimented with resetting the weights on the three-component loss (policy, value, reward) without observing convincing improvements. We did not try to finetune the representation function, the off-the-shelf BERT. A summary of the MuZero hyperparameters configuration is in Table A.4.

Table A.4: Hyperparameters for MuZero.

| Parameter | Value |
|---|---|
| Simulations per Step | 100 |
| Actor Instances | 500 |
| Training TPU Cores | 1 |
| Inference TPU Cores | 7 |
| Initial Inference Batch Size ($h_\theta$) | 4 per core |
| Recurrent Inference Batch Size ($f_\theta$, $g_\theta$) | 32 per core |
| LSTM Units ($g_\theta$) | One layer of 512 |
| Feed-forward Units ($f_\theta$) | One layer of 32 |
| Training Batch Size | 16 |
| Optimizer | SGD |
| Learning Rate | 1e−4 |
| Weight Decay | 1e−5 |
| Discount Factor ($\gamma$) | .9 |
| Unroll Length ($K$) | 5 |
| Max. #actions Expanded per Step | 100 |
| Max. context tokens from document title | 10 |
| Max. context tokens from document content | 70 |

### D.2   T5

The T5 agent is trained for about 5 days on 16 Cloud TPU v3, starting from the pre-trained T5-11B checkpoint. We select the final checkpoint based on the best Dev performance.

**Computational Complexity**   The computational cost of the T5 agent is determined by the T5 model size and sequence lengths. To encode the state we use a maximum sequence length of 512. The decoder predicts the query expansion and has $< 32$ tokens. Additionally we use a beam size of 4. All reported experiments use the largest model, XXL with 11 billion parameters. Smaller models yield competitive but lower results. XXL consists of a 24 layer encoder and decoder with 128-headed attention mechanisms. The "key" and "value" matrices of all attention mechanisms have an inner dimensionality of 128. The feed-forward networks in each block consist of a dense layer with an output dimensionality of 65,536 and all other sub-layers and embeddings have a size of 1024.

**Tuning**   We ran many T5 experiments over the course of the project but didn't perform extensive hyperparameter tuning. As can be seen in Table A.5 we use mostly standard parameters for finetuning the 11B parameter public T5 model following (Raffel et al., 2020). For example we did not experiment with learning-rate schedules, dropout rates, uncommon batch sizes etc. Our experiments mostly explored other design choices: how to represent the input (cf. Table A.3), how much of the context to use (here 30 token context worked slightly better than 70 tokens and was faster to train) and we compared the different grammar types (G0-G4). For these experiments we used the T5-large model because it was quicker and we found that the insights carry over to the larger variants. After training we evaluated several checkpoints and picked the best checkpoint on the Dev set, as is common practice. We then ran the best checkpoint on the test set.

Table A.5: Hyperparameters for T5.

| Parameter | Value |
|---|---:|
| Number of Parameters | 11B |
| Encoder/Decoder Layers | 24 |
| Feed-forward dimension | 65536 |
| KV dimension | 128 |
| Model dimension | 1024 |
| Number of Heads | 128 |
| Batch Size (in tokens) | 65536 |
| Dropout Rate | 0.1 |
| Learning Rate (constant) | 0.0005 |
| Optimizer | AdaFactor |
| Maximum input length (tokens) | 512 |
| Maximum target length (tokens) | 32 |
| Finetuning steps on NQ Train | 41400 |
| Max. context tokens from document title | 10 |
| Max. context tokens from document content | 30 |

## E    Results

Table A.6 reports the results for the different versions of the T5 agent, evaluated on dev. We don't evaluate all agents with the generative answer system, for answer quality we report only the performance of the internal machine reader (EM-MR). Table A.7 reports extended results, including for NQ Dev and the PS/MR component answer quality eval (EM-MR). Moreover, in Figure 4b we plot the performance of our T5-G1 agent on NQ Dev as a function of the maximum number of query refinements. We observed the performance increase monotonically with the number of refinements and that most of the performance gain is achieved in the early steps, in accordance with the respective supervised Rocchio episodes (Figure 4a).

Table A.6: Results of all T5 Agents on NQ Dev.

| Version | NDCG@5 | Top-1 | Top-5 | EM-MR | Reward |
|---|---|---|---|---|---|
| G0 | 40.75 | 52.12 | 64.93 | 30.22 | 33.30 |
| G1 | 43.10 | 52.12 | 66.09 | 29.50 | 35.55 |
| G2 | 41.16 | 51.51 | 63.54 | 30.03 | 33.81 |
| G3 | 41.69 | 51.34 | 64.17 | 29.77 | 33.95 |
| G4 | 41.53 | 50.98 | 63.49 | 29.70 | 34.25 |

Table A.7: Results on NQ Dev and Test.

| Metric | Data | BM25 | +PS | +RM3 | MuZero | T5-G1 | MuZero+T5s | DPR | Rocchio-G4 |
|---|---|---|---|---|---|---|---|---|---|
| NDCG@5 | Dev | 19.83 | 22.95 | 25.09 | 30.76 | 43.10 | 45.30 | - | 64.89 |
| | Test | 21.51 | 24.82 | 26.99 | 32.23 | 44.27 | 46.22 | - | 65.24 |
| Top-1 | Dev | 28.17 | 43.06 | 44.81 | 46.02 | 52.12 | 54.15 | - | 74.99 |
| | Test | 28.67 | 44.93 | 46.13 | 47.97 | 52.60 | 54.29 | 52.47 | 73.74 |
| Top-5 | Dev | 50.47 | 50.47 | 53.61 | 57.71 | 66.09 | 70.05 | - | 88.21 |
| | Test | 53.76 | 53.76 | 56.33 | 59.97 | 66.59 | 71.05 | 72.24 | 88.17 |
| EM-MR | Dev | 15.31 | 25.15 | 26.22 | 27.17 | 29.50 | 31.12 | - | 47.38 |
| | Test | 14.79 | 25.87 | 26.95 | 28.19 | 30.08 | 30.58 | 41.50 | 46.34 |
| EM-T5 | Dev | 28.98 | 40.70 | 41.65 | 32.48 | 44.75 | 44.47 | - | 63.78 |
| | Test | 28.78 | 41.14 | 40.14 | 32.60 | 44.04 | 44.35 | 41.50 | 62.35 |

### E.1    Pseudo-Relevance Feedback Baselines

We investigate the performance of multiple pseudo-relevance feedback (PRF) baselines on our setup. We employ these baselines by running search sessions of length $k$, where, at each step, we choose the *most relevant* term of the top-retrieved documents and add it to the query. To determine the most relevant term, we use either inverse document frequency (IDF), computed over our full retrieval corpus, or RM3 (Jaleel et al., 2004). For RM3, we use the model described in Eq. 20 of Pal et al. (2013) with $\mu = 2500$. After each expansion step, we use the passage scorer (PS) to score and rank the documents. This is an important step, as we do this approach iteratively, so the baseline is more comparable to our agent's setup. While a standard PRF baseline on top of BM25 adds a term to the query (equivalent to our "or"-operator), we investigate the effect of different Lucene operators that our agents have access to. In particular, we run for each of our 10 operators ("or", "+content", "+title", "-content", "-title", "$\wedge$.1", "$\wedge$2", "$\wedge$4", "$\wedge$6", "$\wedge$8") a PRF baseline with $k = 20$ steps (same as our agents). The results are reported in Table A.8. Interestingly, the "-title"-operator, which limits search results not to contain any documents where the specified term is part of the title, works best across all metrics, datasets, and relevancy algorithms. This is in contrast to the standard motivation of PRF to promote relevant terms that appeared in the search results. Instead, *requesting* search results to contain *new* documents (with different titles) seems to be the stronger heuristic. We believe that these experiments underline the benefit of a learned agent to automatically pick the right operator based on the search session context.

Table A.8: Results on NQ Dev and Test for the pseudo-relevance feedback sessions. Here, we run episodes of length 20 where we determine, at each step, the most relevant term from the retrieved results using either inverse-document frequency "IDF" or "RM3". We add the term using one of our 10 operators: simply appending the term ("or"), enforcing the term in the *content* or *title* ("+c"/"+t"), limiting the search to documents that not contain the term in the *context* or *title* ("-c"/"-t"), and boosting the term with different values ("∧.1","∧2","∧4","∧6","∧8"). After each step in the episode, we aggregate the documents using the scores from our passage scorer (PS). The largest value in each table row is indicated in bold, and the second-largest is underlined.

| Metric | Data | Alg | or | +c | +t | -c | -t | ∧.1 | ∧2 | ∧4 | ∧6 | ∧8 |
|--------|------|-----|------|------|------|------|------|------|------|------|------|------|
| **NDCG@5** | Dev | IDF | 24.78 | _25.13_ | 24.61 | 25.12 | **26.81** | 23.67 | 24.45 | 24.43 | 24.37 | 24.30 |
| | Dev | RM3 | 25.09 | _25.41_ | 24.78 | 24.98 | **26.32** | 23.69 | 24.53 | 24.60 | 24.50 | 24.35 |
| | Test | IDF | 26.48 | 26.60 | 26.33 | _27.32_ | **29.33** | 25.51 | 26.35 | 26.25 | 26.19 | 26.08 |
| | Test | RM3 | _26.99_ | 26.98 | 26.70 | 26.90 | **28.59** | 25.47 | 26.60 | 26.61 | 26.54 | 26.37 |
| **Top-1** | Dev | IDF | 44.52 | 44.87 | 44.56 | _45.35_ | **47.09** | 44.13 | 44.45 | 44.36 | 44.30 | 44.21 |
| | Dev | RM3 | 44.81 | 45.21 | 44.45 | _45.56_ | **46.92** | 44.17 | 44.53 | 44.54 | 44.42 | 44.32 |
| | Test | IDF | 45.93 | 45.90 | 46.10 | _47.09_ | **49.29** | 45.84 | 46.18 | 46.01 | 45.98 | 45.90 |
| | Test | RM3 | 46.13 | 46.41 | 46.30 | _47.37_ | **49.03** | 45.78 | 46.41 | 46.24 | 46.18 | 46.10 |
| **Top-5** | Dev | IDF | 53.08 | 53.15 | 52.95 | _54.27_ | **56.49** | 51.74 | 52.59 | 52.68 | 52.61 | 52.58 |
| | Dev | RM3 | 53.61 | 53.85 | 53.19 | _54.29_ | **56.01** | 51.91 | 52.88 | 53.06 | 52.91 | 52.82 |
| | Test | IDF | 55.62 | 55.62 | 55.42 | _57.37_ | **60.14** | 54.96 | 55.56 | 55.50 | 55.50 | 55.42 |
| | Test | RM3 | 56.33 | 56.27 | 56.07 | _57.54_ | **59.58** | 55.04 | 55.99 | 56.02 | 55.99 | 55.93 |

## F  Details and Examples for the Grammar-Guided MCTS

$$
\begin{aligned}
Q &\Rightarrow W\ Q\ |\ U\ Q\ |\ STOP \\
U &\Rightarrow Op\ Field\ W \\
Op &\Rightarrow -\ |\ +\ |\ \wedge_i \\
&\quad i \in \{0.1, 2, 4, 6, 8\} \\
Field &\Rightarrow title\ |\ contents \\
W^x &\Rightarrow V^x\ |\ V^x\ \overline{W}^x \\
\overline{W}^x &\Rightarrow \overline{V}^x\ |\ \overline{V}^x\ \overline{W}^x \\
V^x &\Rightarrow \{w | w \in V_B \wedge \\
&\quad trie(\Sigma^x).HasSubstring(w)\} \\
\overline{V}^x &\Rightarrow \{w | \#w \in V_B^{\#} \wedge \\
&\quad trie(\Sigma^x).HasSubstring(\overrightarrow{w})\}
\end{aligned}
$$

(a) The productions of the query grammar: x identifies a specific vocabulary induced by the aggregated results at time $t$ (index omitted), $V_B$ ($V_B^{\#}$) is the BERT wordpiece prefix (suffix) vocabulary, $\overrightarrow{w}$ denotes the string ending at $w$, including the preceding wordpieces.

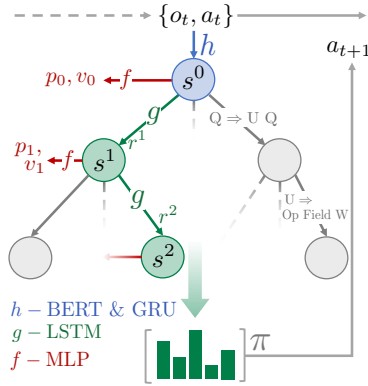

(b) The MuZero MCTS with grammar-guided node expansions represented as edge labelled with CFG rules.

Figure A.3

Figure A.3a lists the detailed rules schemata for the query grammar used by the MuZero agent – explained in Section 3.2.1. An optional STOP rule allows the agent to terminate an episode and return the results collected up to that point. Using the BERT sub-word tokens as vocabulary allows us to generate a large number of words with a total vocabulary size of ∼30k tokens.

Our implementation of MuZero modifies the MCTS to use the query grammar for efficient exploration. Figure A.3b shows the different network components used during MCTS. Each node expansion is associated with a grammar rule. When the simulation phase is complete, the visit counts collected at the children of the MCTS root node provide the policy $\pi$ from which the next action $a_{t+1}$ is sampled.

Each simulation corresponds to one or more hypothetical follow-up queries (or fragments) resulting from the execution of grammar rules. The MCTS procedure executes Depth-First node expansions, guided by the grammar, to generate a query top-down, left-to-right, in a forward pass. To control the process, we add two data structures to MCTS nodes: a stack $\gamma$, and an output buffer $\omega$: $\gamma$ contains a list of unfinished non-terminals, $\omega$ stores the new expansion. The stack is initialized with the start symbol $\gamma = [Q]$. The output buffer is reset, $\omega = []$, after each document search. When expanding a node, the non-terminal symbol on the top of $\gamma$ is popped, providing the left-hand side of the rule associated with the new edge. Then, symbols on the right-hand side of the rule are pushed right-to-left onto $\gamma$. When a terminal rule is applied, the terminal symbol is added to $\omega$. The next time $\gamma$ contains only Q, $\omega$ holds the new query expansion term $\Delta q_t$ to be appended to the previous query $q_t$ for search.

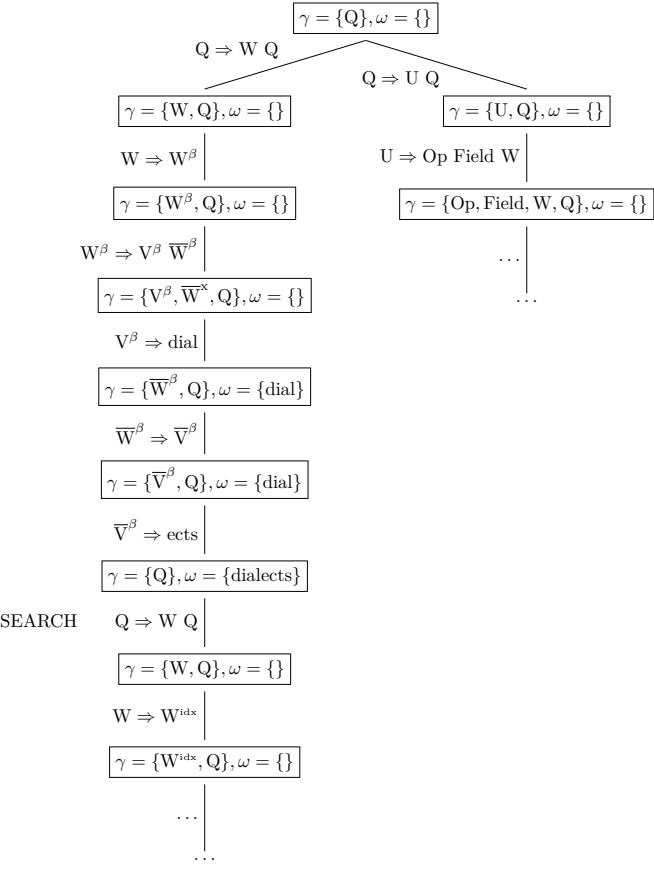

Figure A.4

Figure A.4 illustrates the process. Nodes represent the stack $\gamma$ and output buffer $\omega$. Edges are annotated with the rule used to expand the node. We illustrate the left-branching expansion. Starting from the top, the symbol "Q" is popped from the stack, and a compatible rule, "Q $\Rightarrow$ W Q", is sampled. The symbols "W" and "Q" are added to the stack for later processing. The agent expands the next node choosing to use the document content vocabulary (W $\Rightarrow$ W$^\beta$), then it selects a vocabulary prefix ('dial'), adding it to the output buffer $\omega$, followed by a vocabulary suffix ('ects'). At that point, the stack contains only Q, and the content of $\omega$ contains a new expansion, the term 'dialects'. A latent search step is simulated through MuZero's $g_\theta$ sub-network. Then the output buffer $\omega$ is reset.

After the latent search step, the simulation is forced to use the full trie ($W \Rightarrow W^{\text{idx}}$), which includes all terms in the Lucene index. This is necessary since there is no observable context that can be used to restrict the vocabulary. Instead of simply adding an OR term ($Q \Rightarrow W\ Q$), the right branch of the example selects an expansion with unary operator and field information ($Q \Rightarrow U\ Q$).

## G    Search Session Examples

Table A.9 and Table A.10 show example Rocchio sessions using the full grammar. Table A.11 shows a session generated by the MuZero agent.

Table A.9: Example episode from a Rocchio session with grammar G4 (Rocchio-G4). The question asks for the name of the "green guy from sesame street" (referring to "Oscar the Grouch", a green *muppet* that lives in a *trash* can on Sesame street). The query expansions add the requirement that the content of the documents should contain the words "muppet", and "trash"; both terms closely related to the answer "Oscar the Grouch" but not mentioned in the original query. The score increases from 0.040 for the original query to 0.891 for the final query.

|  | Query and Search Results | Score |
|---|---|---|
| $q_0$ | who is the green guy from sesame street | |
| | Top-2 documents retrieved with $q_0$: | 0.040 |
| $d_1$ | Title   Music of Sesame Street
Content   . . . **Christopher Cerf**, who Gikow called "the go-to guy on "Sesame Street" for classic rock and roll as well as song spoofs . . . | |
| $d_2$ | Title   Sesame Street characters
Content   . . . **Forgetful Jones**, a "simpleton cowboy" with a short-term memory disorder; and even Kermit the Frog, the flagship character of The Muppets. . . . | |
| $q_1$ | who is the green guy from sesame street +(contents:"muppet") | |
| | Top-2 documents retrieved with $q_1$: | 0.505 |
| $d_1$ | Title   History of Sesame Street
Content   . . . Raposo's "I Love Trash", written for . . . **Oscar the Grouch**, was included on the first album of "Sesame Street" songs, . . . | |
| $d_2$ | Title   Julie on Sesame Street)
Content   . . . Andrews and "special guest star" Como interacted with the Muppet characters (including Kermit the Frog, Big Bird, Cookie Monster, . . . **Oscar the Grouch** and Bert and Ernie), . . . | |
| $q_2$ | who is the green guy from sesame street +(contents:"muppet") +(contents:"trash") | |
| | Top-2 documents retrieved with $q_2$: | 0.891 |
| $d_1$ | Title   A Muppet Family Christmas
Content   . . . all the Muppets sing a medley of carols and swap presents (except **Oscar the Grouch**, who just sits in his trash can, sighing very miserably due to his hatred for Christmas). . . . | |
| $d_2$ | Title   Music of Sesame Street
Content   . . . He wrote "I Love Trash" for **Oscar the Grouch**, which was included on the first album of "Sesame Street" songs. . . . | |

Table A.10: Example of a Rocchio session with grammar G4 (all terms).

| | Query and Search Results | Score |
|---|---|---|
| $\mathbf{q}_0$ | who were the judges on the x factor | 0.043 |
| $\mathbf{d}_1$ | Title    The X Factor (Australian TV series) 
 Content   ... After "The X Factor" was revived for a second season in 2010, **Natalie Garonzi** became the new host of "The Xtra Factor" on ... | |
| $\mathbf{d}_2$ | Title    X Factor (Icelandic TV series) 
 Content   ... The judges were the talent agent and businessman Einar Bárðarson, rock musician **Elínborg Halldórsdóttir** and pop singer Paul Oscar ... | |
| $\mathbf{q}_1$ | who were the judges on the x factor (contents:"confirmed"∧4) | 0.551 |
| $\mathbf{d}_1$ | Title    The X Factor (U.S. season 2) 
 Content   ... **Simon Cowell** and L.A. Reid returned as judges, while Paula Abdul and Nicole Scherzinger were replaced ... | |
| $\mathbf{d}_2$ | Title    The X Factor (New Zealand series 1) 
 Content   ... "The X Factor" was created by **Simon Cowell** in the United Kingdom and the New Zealand version is based on ... | |
| $\mathbf{q}_2$ | who were the judges on the x factor (contents:"confirmed"∧4) (title:"2"∧8) | 0.678 |
| $\mathbf{d}_4$ | Title    The X Factor (U.S. season 2) 
 Content   ... It was also reported that Cowell was in talks with **Britney Spears** for her to join the show, ... | |
| $\mathbf{d}_5$ | Title    H.F.M. 2 (The Hunger for More 2) 
 Content   ... Confirmed guests include **Eminem, Kanye West**, Lloyd, Juelz Santana, 50 Cent, Styles P, ... | |
| $\mathbf{q}_3$ | who were the judges on the x factor (contents:"confirmed"∧4) (title:"2"∧8) +(contents:"britney") | 0.804 |
| $\mathbf{d}_3$ | Title    The X Factor (U.S. TV series) 
 Content   ... Reid, former "The X Factor" judge Cheryl Cole, and Cowell's former "American Idol" colleague **Paula Abdul** were confirmed to join Cowell in the judging panel ... | |
| $\mathbf{d}_5$ | Title    The X Factor (U.S. season 2) 
 Content   ... It was also reported that Cowell was in talks with **Britney Spears** for her to join the show, ... | |
| $\mathbf{q}_4$ | who were the judges on the x factor (contents:"confirmed"∧4) (title:"2"∧8) +(contents:"britney") (contents:"cowell"∧4) | 0.926 |
| $\mathbf{d}_1$ | Title    The X Factor (U.S. season 2)      (BM25 Rank: 15) 
 Content   ... **Simon Cowell** and L.A. Reid returned as judges, while Paula Abdul and Nicole Scherzinger were replaced ... | |
| $\mathbf{d}_2$ | Title    The X Factor (New Zealand series 1)      (BM25 Rank: 195) 
 Content   ... "The X Factor" was created by **Simon Cowell** in the United Kingdom and the New Zealand version is based on ... | |
| $\mathbf{d}_3$ | Title    Louis Walsh      (BM25 Rank: >1000) 
 Content   ... He joined the other season two judges: L.A. Reid, Demi Lovato and **Britney Spears**, and was introduced with the line, ... | |
| $\mathbf{d}_4$ | Title    The X Factor (U.S. TV series)      (BM25 Rank: 206) 
 Content   ... Reid, former "The X Factor" judge Cheryl Cole, and Cowell's former "American Idol" colleague **Paula Abdul** were confirmed to join Cowell in the judging panel ... | |
| $\mathbf{d}_5$ | Title    Simon Cowell      (BM25 Rank: >1000) 
 Content   ... Cowell and Reid returned for season 2, while Demi Lovato and **Britney Spears** joined the judging panel as replacements for Abdul and Scherzinger ... | |

Table A.11: Example of a MuZero agent search session.

| | Query and Search Results | Score |
|---|---|---|
| $\mathbf{q}_0$ | who won the wwe money in the bank | 0.071 |
| $\mathbf{d}_1$ | Title      Money in the Bank (2017) 
 Contents   . . . In the main event, Baron Corbin won the men's ladder match, earning a contract for a WWE Championship match, while **Carmella** controversially won the first women's ladder match to earn a SmackDown Women's Championship match contract . . . | |
| $\mathbf{q}_1$ | who won the wwe money in the bank (contents:"jinder"∧2) | 0.130 |
| $\mathbf{d}_1$ | Title      Money in the Bank ladder match 
 Contents   . . . For the traditional ladder match, which had a contract for a match for Smack-Down's WWE Championship, SmackDown Commissioner Shane McMahon announced AJ Styles, Shinsuke Nakamura, Dolph Ziggler, Sami Zayn, and **Baron Corbin** as the original five participants . . . | |
| $\mathbf{q}_4$ | who won the wwe money in the bank (contents:"jinder"∧2) (contents:"dolph"∧2) (contents:"won"∧2) (contents:"zayn"∧2) | 0.414 |
| $\mathbf{d}_1$ | Title      Money in the Bank (2018) 
 Contents   . . . At Backlash, Lashley and **Braun Strowman** defeated Kevin Owens and Sami Zayn. During an interview on the May 7 episode, Lashley spoke . . . | |
| $\mathbf{q}_5$ | who won the wwe money in the bank (contents:"jinder"∧2) (contents:"dolph"∧2) (contents:"won"∧2) (contents:"zayn"∧2) (contents:"strowman"∧2) | 0.587 |
| $\mathbf{d}_2$ | Title      Kevin Owens 
 Contents   . . . Later that night, Owens teaming up with Zayn, The Miz, Curtis Axel and Bo Dallas and lost to Finn Bálor, Seth Rollins, **Braun Strowman**, Bobby Lashley and Bobby Roode in a 10-man tag team match . . . | |
| $\mathbf{q}_7$ | who won the wwe money in the bank (contents:"jinder"∧2) (contents:"dolph"∧2) (contents:"won"∧2) (contents:"zayn"∧2) (contents:"strowman"∧2) (contents:"first"∧2) (contents:"roode"∧2) | 0.848 |
| $\mathbf{d}_1$ | Title      Bobby Lashley          (BM25 Rank: >1000) 
 Contents   . . . Lashley participated in the Greatest Royal Rumble at the namesake event, entering at #44 and scoring two eliminations, but was eliminated by **Braun Strowman**. The first month of Lashley's return would see him in a number of tag-team matches, . . . | |
| $\mathbf{d}_2$ | Title      Kevin Owens and Sami Zayn          (BM25 Rank: >1000) 
 Contents   . . . Later that night, Owens teaming up with Zayn, The Miz, Curtis Axel and Bo Dallas and lost to Finn Bálor, Seth Rollins, **Braun Strowman**, Bobby Lashley and Bobby Roode in a 10-man tag team match . . . | |
| $\mathbf{d}_3$ | Title      Money in the Bank (2018)          (BM25 Rank: 282) 
 Contents   . . . At Backlash, Lashley and **Braun Strowman** defeated Kevin Owens and Sami Zayn. During an interview on the May 7 episode, Lashley spoke . . . | |

