# OpenReview forum: "Boosting Search Engines with Interactive Agents"
_TMLR — Accepted by TMLR_

### Review · Reviewer_CJo8 · 2022-04-11

**Summary Of Contributions:**

The paper presents the role of interactive agents in the search domain and opens up the research. First, the role of self-supervised
language models with inbuilt knowledge about domain vs. the role of RL-based agents that do not have domain knowledge. The experiments
are conducted in both settings and results show that RL-based agents provide better search policies compared to LM models. The experiments are conducted in a SOTA Open-QA environment. The paper concludes that LM and RL agents can be effective in the search domain.

**Broader Impact Concerns:**

Nothing as far I know.

**Requested Changes:**

1. I would suggest mentioning Rocchio query expansion for best/worst-case scenarios (tree depth, time/space complexity)
2. Any hyper-parameter tuning is done?

**Strengths And Weaknesses:**

Strengths
- Good step in advancing search with the involvement of agents
- Experiments are clear.
- Datasets look ideal

Weaknesses
- Suggest authors mention computation complexity of both LM, RL agents

---

> ### Author Response · Authors · 2022-04-27
> **Response to Reviewer CJo8**
>
>
> We thank the reviewer for their positive appreciation of the paper and the helpful comments for changes that we incorporated into the revised version of the paper.
>
>
> ## Weaknesses 1 - Computational Complexity
> In the following a discussion about the computational complexity that we added to Appendix B (State Construction) and Appendix D (MuZero and T5) of the revised version of the paper.
>
> #### State construction
> Both agents expect an input constructed from a set of paragraphs centered around a predicted answer of a machine reader system. Hence, the state construction involves running a BERT-base (110M parameters) machine reader separately (albeit possibly in parallel) over five passages.
>
> #### MuZero
> The computational complexity of a single step of our MuZero agent is determined by the complexity of the state encoding function ("h" in Figure A.3 b) and the number of simulations during MCTS. For the state encoding function, we use BERT-base (110M parameters) to encode the state and a GRU cell with a hidden size of 32 to encode the past actions. The maximum sequence length of the state is 512. The recurrent inference function during the MCTS ("g" in Figure A.3 b) is an LSTM with a hidden dimension of 512 that is invoked num_simulations (typically 100) times. On top of the LSTM representation, we use MLPs ("f" in Figure A.3 b) with a single hidden layer with hidden dimension 512 as the policy, value, and reward head.
>
> #### T5
> The computational cost of the T5 agent is determined by the T5 model size and sequence lengths. To encode the state we use a maximum sequence length of 512. The decoder predicts the query expansion and has <32 tokens. Additionally we use a beam size of 4. All reported experiments use the largest model, XXL with 11 billion parameters. Smaller models yield competitive but lower results. XXL consists of a 24 layer encoder and decoder with 128-headed attention mechanisms. The “key” and “value” matrices of all attention mechanisms have an inner dimensionality of 128. The feed-forward networks in each block consist of a dense layer with an output dimensionality of 65,536 and all other sub-layers and embeddings have a size of 1024.
>
> ## Requested Changes 1 - Rocchio Query Expansion Analysis
> We updated the paper providing more details on the complexity of the synthetic Rocchio query expansions, together with suitable plots that illustrate informative statistics on the resulting Rocchio search sessions. These are provided in Section 5.1 and summarized here for completeness:
>
> Given the original query, we attempt at most 100 possible refinements for each grammar operator. For instance, for the ‘+’ operator we attempt refinements of the form ‘+(field: term)’, where term is taken from the top-100 terms in the intersection dictionary $\Sigma^{\uparrow}_t$ (see Eq. (3) in the paper) and field represents the field (content or title)  where such term was found. By letting G represent the number of available grammar operators (i.e., ‘+’, ‘-’, etc.), this corresponds to issuing 100\*G queries to BM25. Based on these, we select the refinement leading to the highest gain in score (computed according to Eq. (7) in the paper) and neglect the other ones. Then, this process continues until no score-improving refinement can be found, for a maximum of 20 refinement steps. Thus, the overall Rocchio expansion performs at most 20\*100\*G queries to BM25. However, as also explained in Section 5.1,  most Rocchio expansions terminate after a number of steps significantly smaller than 20, either because the maximum score is reached or because no score improvements can be found. Moreover, we have observed the first query expansion steps produce much higher score gains with respect to later ones. This indicates that performing longer Rocchio expansions has diminishing marginal gains. To support these observations, in Section 5.1 we have provided the following two plots: a histogram which shows the length of Rocchio query expansions for different grammars on NQ Dev, and a plot indicating the average score gain resulting from each Rocchio expansion step. Moreover, in Table A.1 we have reported the total number of Rocchio expansion steps on NQ Train for the different grammars, which constitute the training data for our T5 agents.
>
> ## Requested Changes 2 - Hyperparameter Tuning
> We added one paragraph for each agent architecture (MuZero, T5) to Appendix D.1 and D.2 describing the tuning process in detail.

---

### Review · Reviewer_7tKc · 2022-04-13

**Summary Of Contributions:**

When searching for something simple, it's possible to find the information one is looking for easily. However, when searching for something more complex or detailed, searches are often iterative in that the results are assessed and another search is conducted. This work focuses on this problem and focuses on how to design search agents that are able to iteratively refine search queries for question answering. To enable this, the authors leverage recent advances in language modeling and also explore reinforcement learning. It also includes a self-supervised methodology for training such an agent through synthetic search sessions that mimic human users --- this utilizes rocchio query expansion. At a high level, the authors leverage a set of question, answer pairs and iteratively refine the query based on a reward function operating on selecting the document containing the answer.

**Broader Impact Concerns:**

n/a the authors lightly address this and I find it sufficient

**Requested Changes:**

- The construction of the work is that the search engine is the environment and the agent simulates the user. It would be good to reflect on how this simulation reflects reality: for a human, how many times are they refining these queries and how much content are they looking at? The agent learns many policies, but are any of them efficient/reflective of how people use search engines? I believe this is something that is required to contextualize the work.
- Please consider some diagram, adding more captions, etc to make the paper easier to read and follow. I believe this is critical for acceptance.
- I would provide more insight into the different policies the agent is learning and analyzing the performance. There is briefly discussion on this on page 7, but there are also many interesting possible results (some appear in the Appendix). This would make the work stronger. It could also be interesting to reflect on the OpenQA-NQ dataset and if limitations exist for this task.

**Strengths And Weaknesses:**

Strengths:
- It is a very interesting problem space, and the authors outline & motivate the problem clearly.
- The authors highlight several downsides of needing to scale live search, and the ideas of how to iteratively refine through simulated queries enables machine learning.

Weaknesses:
- The introduction of the paper is far more clear than the body. The authors could do a better job of structuring the rest of the article and how the pieces fit together. Further, some bits of the paper are more difficult to read than needed. A good example is Table 1, where there are a large number of abbreviations in one table. I would also provide a lot more concrete examples of these search queries and how they are refined to explain sections such as 3.1.
- It is not clear to me how many times iterative refinement is really needed on the OpenQA dataset. For an average human, how many times does it take to refine the query and find the answer? Is it really necessary to explore the documents so deeply (see Figure 2b)? Based on the results, this exploration is not necessary. It would have been interesting to explore results when capping the depth.

---

> ### Author Response · Authors · 2022-04-27
> **Response to Reviewer 7tKc**
>
> We thank the reviewer for their helpful feedback, especially regarding the presentation of the work. We updated the paper with a particular focus on readability, analysis, and search session insights. In the following, we address all the points individually.
>
> ## Weaknesses 1 and Requested Changes 2 - Presentation
> We uploaded a new version of the paper that improves the readability and presentation, including tables, examples, and captions. In particular,
> * We simplify Figure 1 and present an actual example episode from the Rocchio policy. We describe in detail the full episode (including showing the retrieval results step-by-step) in Table A.8 in the Appendix.
> * We add an additional observed example in Table 1 showcasing different operator usage at the beginning of the paper to make the setup more graspable for the reader early on.
> * We revise and extend the captions of the former Figure 1, Table 1, Table 2, and Figure 2, focusing on understandability.
> * We add a qualitative analysis of the T5-G1 (boosting only) vs. the T5-G4 (all operators) using an episode example in Table 4.
> * We explain the setup for obtaining the Rocchio session data in more detail (Sec. 5.1).
> * We add further analysis of the Rocchio session data (Figure 2) and compare the step-wise performance metrics of the best Rocchio session data vs. the best learned T5 agent (Figure 4).
>
> ## Weaknesses 2 - Iterative Refinements
> * We have computed a histogram of how many steps are executed in the Rocchio Expansion sequences on OpenQA-NQ Dev, and the cumulative score increases at each step. The new plots are provided in the updated paper in Figure 2. In a nutshell: the mean number of steps before the procedure stops because no further gains are possible, for all grammars, is between 4 and 5. Thus, it is beneficial to perform several steps. At the same time, the residual reward drops fast; most of the gains are achieved in the first 2 to 3 steps. Furthermore, we add two plots in Figure 4 showing the average performance of the Rocchio session and the T5-G1 agent per refinement step.
> * We want to clarify that the "depth" shown in former Figure 2b (3b of the revised version) shows the retrieval rank that the documents chosen by the agents would have, based on the original query and the LuceneBM25 document ranking. To put it more simply, how deep would we need to look with the original question to retrieve the documents we find with the agents? We add a comprehensive caption to the figure to make this clear. The histogram of Figure 2b (now 3b) shows that a large portion of documents returned by the agents only appears beyond rank 1000 of the original question, underlining the agents' capability to uncover previously unreachable evidence. As a qualitative example, we refer the reader to the episodes of Table A.9 and A.10; here, we show the BM25 rank of the final documents for the original question. Again, we see that the agents find useful evidence buried very deep in the retrieval results of the original query. Exploration (and recall) is one of the key challenges in IR. Here we want to highlight that the development of ML-based search agents is a promising direction, complementary to the retrieval engine, towards helping users by bringing evidence that would be otherwise out of reach to the very top of the results within the user's attention scope.
>
> ## Requested Changes 1 - Artificial vs. Human Search Policies
> We thank the reviewer for this request to contextualize the search policies with respect to human search. We added a detailed discussion to the revised version of the paper in Section 2 and Section 5.4.
>
> ## Requested Changes 3 - Policy Analysis / OpenQA-NQ Limitations
> * We extend the quantitative and qualitative analysis to provide more insights into the search episodes by making the following changes to the paper:
>   * We provide two additional plots analyzing the Rocchio sessions (Figure 2). One histogram analyzes the Rocchio sessions’ length for different grammars, and the second plot shows the average score gain per step.
>   * We add two plots (Figure 4) showing the average performance per query refinement for the Rocchio-G4 sessions and the T5-G1 agent episodes.
>   * We add a qualitative analysis of the T5-G1 (boosting only) vs. the T5-G4 (all operators) using an episode example in Table 4.
>   * We provide two additional Rocchio query expansion episodes (one in the main body of the paper and another one in the Appendix), including the top retrieved results and the documents' score. We provide a detailed analysis of those in the respective captions for clarity.
>   * We extend the discussion of the analysis plots in former Figure 2 (now Figure 3) by providing a detailed caption.
> * We add a paragraph “Thoughts on OpenQA-NQ” in Section 5.4 of the revised paper to address the limitations of OpenQA-NQ.

---

### Review · Reviewer_28ro · 2022-04-22

**Summary Of Contributions:**

The paper presents an imitation learning as well as a reinforcement learning approach to design agents that can interact with search engines. The agent refines an initial search query, mostly through query expansion and a list of available operators such as term weighting. The documents resulting from the refined query are used to produce an answer for the original query.

A Rocchio expansion-based policy is used to generate training data for a T5-based imitation learning agent. The agent is trained in a text-to-text manner from the original query to a reformulated one. The MuZero agent explores the search environment with MCTS that is guided by a hand-defined CFG. The per action reward for the agent is given by a combination of 1) NDCG at 5 2) a variant of NDCG where a document is weighted highly if the reader network is able to extract the correct answer from it and 3) Passage scores.

The results show that the T5-based agent is able to outperform a BM25 and is close to a strong DPR baseline. The MuZero approach by itself doesn't do particularly well, but when ensembles with the T5 agent, is often able to outperform DPR on retrieval as well as finding the answer span.

**Requested Changes:**

1. The Pseudocode/algorithm for the Rocchio expansion (Algo 1 in the appendix) seems critical to the understanding of this paper. I would recommend bringing it to the main body of the paper, subject to space constraints. This is not critical, but would likely give readers a better understanding of Section 2.2.
2. As mentioned in Point 5 of the previous section, the paper will benefit from some statistics about the data collected from Rocchio search sessions. This doesn't have to be in the main body of the paper.
3. Reiterating point 4 in the section above, a plot showing the answer EM/NDCG as a function of the number of query refinements will strengthen this paper. This again doesn't have to be in the main body of the paper.

**Strengths And Weaknesses:**

Strengths:

1. The paper is very well written and the need for an agent that can interact with a search engine to refine a user's search query is well motivated.
2. The idea of using Rocchio expansions to generate imitation learning data is clever and I think there is some opportunity for better environment exploration with data augmentation of the initial or intermediate queries in future work.
3. The T5 agent shows impressive improvements over BM25 as well as re-ranking BM25 with passage scores and ensembling it with the MuZero agent makes it competitive with DPR (especially in terms of precision).

Weaknesses:
1. While it is commendable that the authors were able to get an RL agent to work with such large action spaces and sparse rewards, the results overall underperform when compared to their simpler T5 baseline, based on behavior cloning.

Questions & Comments:
1. Could you consider adding noise to q_0 in your offline data generation process? This noise function could be something that drops n-grams or inserts random ones. The idea here is to potentially broaden the set of training data you’d obtain from Rocchio Expansions and have the model learn to add/remove terms often instead of mostly relying on term weighting. This could also help with ill-specified user queries/typos etc.
2. The above noising approach and in particular, adding/removing n-grams to q_0, may help the MuZero policy to learn to also potentially add/remove n-grams instead of just learning to reweight, which as you mention the agent seems to default to.
3. Can the final RC model be conditioned on the refined search query as well as the original to better extract the answer? This will likely require finetuning of the RC model to parse special symbols for weighting etc, but given you have supervised data like (D, q0, qt, answer), this seems possible.
4. Do you use techniques to avoid something like "exposure bias" with behavior cloning or are search sessions small enough that this isn't a problem in practice? Does the answer EM monotonically increase as you keep refining your query? It would be interesting for readers to see a plot of answer EM or NDCG as a function of the number of query refinements.
5. The paper seems to lack some detail on the nature of the final data collected from Rocchio search sessions. 1) How many training examples are available at the end? 2) What is the average search session length? 3) Does a query have multiple search sessions associated with it?

---

> ### Author Response · Authors · 2022-04-27
> **Response to Reviewer 28ro**
>
> We thank the reviewer for their insightful comments, improvement points for the paper, and further experimental suggestions.
>
> ## Requested Changes 1 & 2 and Comment 5 - Rocchio Sessions
> We recognize, also in accordance with the other reviews, that the paper could benefit from additional details on the synthetic Rocchio sessions introduced in Section 2. Hence, we have provided in-depth details and statistics about these in an updated version of the paper.
> In particular, in our new Section 5.1 we have:
> * provided a comprehensive description of a prototypical Rocchio search session.
> * added a histogram which shows the length of Rocchio query expansions for different grammars on NQ Dev. The histogram shows that most Rocchio expansions terminate after a number of steps significantly smaller than 20 (which is our hard maximum), either because the maximum score is reached or because no score improvements can be found.
> * provided a plot showing the average score gain resulting from each Rocchio expansion step. The plot shows the first steps produce much higher score gains with respect to later ones, and thus that performing longer Rocchio expansions has diminishing marginal gains.
>
> We considered moving Algorithm 1 to the main body of the paper. However, in light of the increasing number of figures and examples in the main body and the size of the algorithm description, we opted for referencing the algorithm prominently in Sec. 5.1 while keeping it in the appendix.
>
> Regarding additional statistics about the collected Rocchio sessions, we added Table A.1, where we report the total number of Rocchio expansion steps on NQ Train for the different grammars, which constitute the training data for our T5 agents.
>
>
> ## Requested Changes 3 and Comment 4 - Per-Step Performance / Exposure Bias
> We provide plots for the Rocchio-G4 session and the T5-G1 agent showing the performance per step in Figure 4 of the revised paper. We see that for both the Rocchio session "training" data and the T5 agent's actual episodes, the retrieval (NDCG, Top-k) and the reader metrics (EM) increase close to monotonic with the number of steps. Consistent with the plots in Figure 2, most of the performance gain is achieved within the first steps, while such gains are negligible after 20 steps. Intuitively, this sort of robustness can be explained by using the PS score to keep the top-k documents observed so far. This ensures that our agents can alter the set of top-k documents only if the selected refinements bring up documents with a high PS score.
>
> Regarding exposure bias, we agree that this is an important direction that we see developing along the line of work on RL-based T5 agents mentioned in the second paragraph of Section 5.4. In several experiments, we have evaluated 1) sampling positive and negative Rocchio sessions, 2) mixing MuZero and Rocchio sessions data, and 3) sampling additional training data from a trained T5 search agent to continue fine-tuning the same model. None of these experiments has improved performance so far, but these are only preliminary investigations.
>
>
> ## Comment 1 & 2 - Experiments with added Noise
> We thank the reviewer for this interesting idea for generating additional data that might lead to more diverse policies. Even though we thought about a variant of such an approach during the early phase of this project, we have not experimented with changing the initial queries, neither during training nor during inference. We agree that there is potential to generate better search episodes using data augmentation/adaption and we think this is an excellent direction for future work.
>
> ## Comment 3 - Train Reader with Refinements
> We already did experiments in the proposed direction without experiencing performance gains. We experimented with training the T5 answer generation component using the full query at each step, including the augmentation operations. The model received the original query, the refinements done up to the point, and the top 5 documents as input. It was trained to predict the answer to the question and the subsequent refinement. Preliminary evaluations showed that the model was performing close to the agent models but significantly poorer than the simpler answer generation models. We hypothesized that including the refinements in the state allowed less information from documents to be incorporated due to input size constraints. The state string grows quickly with additional refinements represented, pushing out content from the observed documents.

---

### Decision · Action_Editors · 2022-05-22

**Recommendation:** Accept with minor revision

**Comment:**

The paper tackles the problem of designing an agent that can interact with a search engine via a predefined query language. The paper presents two methods for doing so, an imitation learning baseline built on top of T5 and a MuZero agent that explores the space of reformulations.

All reviewers found the problem the paper is tackling interesting and timely. The authors have provided a much improved post-rebuttal version, including suggestions from the reviewers. As some reviewers pointed out, one of the weaknesses of the paper is that the MuZero agent generally under-performs the behavioural cloning T5 method. Another weakness I can point out is that the paper lacks a simple pseudo-relevance feedback baseline, which also exploits multiple retrieval phases where the query is built in a rather simple manner. Nevertheless, all reviewers lean for acceptance. I concur with reviewers.

Even if the results in this paper aren't particularly strong, the general problem tackled by this paper and the ideas therein contained will be of interest to TMLR's audience. I think it is in the spirit of TMLR to favour papers that explore interesting research areas even without particularly strong results. I found interesting the idea of exploiting Rocchio expansions to guide the T5 behavioral cloning baseline. The extent of the analyses and experiments performed puts this paper slightly above the bar for acceptance.

I recently suggested the authors to implement a pseudo-relevance feedback baseline. I do not believe the results of the PRF baseline can invalidate results and methods used in this paper. Therefore, I recommend "Accept with minor revision".

---

> ### Author Response · Authors · 2022-05-25
> **Thank you!**
>
> We thank the action editor and all reviewers for their positive appreciation of the paper! Moreover, we would like to thank them for their helpful comments and suggestions that have strongly improved the paper throughout the reviewing process.
>
> We've just uploaded a revised version of the paper that adds the requested PRF baseline in Table 2 and further studies different PRF variants in Section E.1 of the Appendix.